

# Spatial and temporal patterns of sediment storage and erosion following a wildfire and extreme flood

Daniel J. Brogan[1], Peter A. Nelson[1], and Lee H. MacDonald[2]

[1]Department of Civil and Environmental Engineering, Colorado State University, Fort Collins, Colorado 80523-1372, USA
[2]Department of Ecosystem Science and Sustainability, Colorado State University, Fort Collins, Colorado 80523-1476, USA

**Correspondence:** Peter A. Nelson (peter.nelson@colostate.edu)

**Abstract.** Post-wildfire landscapes are highly susceptible to rapid geomorphic changes at both the hillslope and watershed scales due to increases in hillslope runoff and erosion, and the resulting downstream effects. Numerous studies have documented these changes at the hillslope scale, but relatively few studies have documented larger-scale post-fire geomorphic changes over time. In this study we used five airborne laser scanning (ALS) datasets collected over four years to quantify valley bottom changes in two ~15 km$^2$ watersheds, Skin Gulch and Hill Gulch, after the June 2012 High Park fire in northern Colorado and a large mesoscale flood 15 months later. The objectives were to: 1) quantify spatial and temporal patterns of erosion and deposition throughout the channel network following the wildfire and including the mesoscale flood; and 2) evaluate whether these changes are correlated to precipitation metrics, burn severity, or morphologic variables. Geomorphic changes were calculated using a DEMs of difference (DoD) approach for the channel network segmented into 50-m lengths. The results showed net sediment accumulation after the wildfire in the valley bottoms of both watersheds, with the greatest accumulations in the first two years after burning in wider and flatter valley bottoms. In contrast, the mesoscale flood caused large net erosion, with the greatest erosion in the areas with the greatest post-fire deposition. Volume changes for the different time periods were weakly but significantly correlated to, in order of decreasing correlation, contributing area, channel width, percent burned at high and/or moderate severity, channel slope, confinement ratio, maximum 30-minute rainfall, and total rainfall. These results suggest that morphometric characteristics, when combined with burn severity and a specified storm, can indicate the relative likelihood and locations for post-fire erosion and deposition. This information can help assess downstream risks and prioritize areas for post-fire hillslope rehabilitation treatments.

*Copyright statement.* TEXT

# 1 Introduction

Wildfires alter hydrologic response by creating conditions that can lead to greatly increased runoff and erosion rates. At plot to hillslope scales increased rates of runoff have been attributed to a decrease in canopy cover, ground cover and surface roughness, and an increase in soil sealing and soil water repellency (e.g., Benavides-Solorio and MacDonald, 2001; Huffman et al., 2001;



Larsen and MacDonald, 2007; Onda et al., 2008; Larsen et al., 2009; Ebel et al., 2012; Stoof et al., 2012; Schmeer et al., 2018). At the hillslope scale these fire-induced changes increase a variety of erosional processes, including rainsplash, sheetflow, rilling, gullying, landslides, and debris flows (e.g., Benda and Dunne, 1997; Inbar et al., 1998; Cannon et al., 2001; Gabet and Dunne, 2003; Roering and Gerber, 2005; Wagenbrenner and Robichaud, 2014; Rengers et al., 2016). As spatial scale increases

channel erosion can become important (e.g., Meyer et al., 1992; Legleiter et al., 2003; Wagenbrenner and Robichaud, 2014), but the literature predominantly reports post-fire deposition, including alluvial fans, channel infilling, floodplain accretion, reservoir filling, and a sediment superslug (e.g., Moody and Martin, 2001; Reneau et al., 2007; Santi et al., 2008; Orem and Pelletier, 2015; Moody, 2017).

Considerable advances have been made in understanding post-wildfire runoff, erosion, and mass wasting at hillslope and

small watershed scales (see Shakesby and Doerr, 2006; Moody et al., 2013, and references within); however, the larger-scale effects of fires on flooding, water quality, and sedimentation are often the most significant due to their adverse human and resource impacts (Hamilton et al., 1954; Doehring, 1968; Moody and Martin, 2001, 2004; Rhoades et al., 2011; Writer et al., 2014). Most efforts to model post-fire runoff and erosion have focused at the hillslope scale, and include WEPP (e.g., Elliot, 2004; Miller et al., 2011), RUSLE (Renard et al., 1997), AGWA (Goodrich et al., 2005), and ERMiT (Robichaud et al., 2007).

The first two models have been used as the basic building blocks for predicting changes at scales larger than a few hundred hectares (e.g., GeoWEPP; Miller et al., 2011; Elliot et al., 2016), but downstream post-fire flooding, erosion, and sedimentation are not a simple sum of hillslope-scale processes. Accurate predictions and upscaling from hillslopes require a more explicit consideration of sediment storage and erosion, and a failure to do so will result in unreliable estimates of watershed-scale peak flows, sediment production, sediment deposition, and sediment delivery (e.g., Moody and Kinner, 2006; Stoof et al.,

2012). Larger-scale studies also have generally quantified sediment delivery rather than explicitly evaluating the magnitude and controls on the spatially-varying geomorphic changes over the channel network (e.g., Pelletier and Orem, 2014; Orem and Pelletier, 2015). Efforts to measure and better understand these larger-scale geomorphic changes have been hampered by the lack of high spatial- and temporal-resolution data at the watershed scale (Moody et al., 2013).

To some extent the larger-scale effects of fires should be analogous to the observed patterns of erosion and deposition

following large floods (e.g., Wolman and Eiler, 1958). More specifically, stream power—or gradients in stream power—and lateral confinement have generally been the best predictors of the spatial patterns of erosion and deposition (e.g., Miller, 1995; Fuller, 2008; Thompson and Croke, 2013; Gartner et al., 2015; Stoffel et al., 2016; Surian et al., 2016; Yochum et al., 2017), although strong correlations are not always apparent (e.g., Nardi and Rinaldi, 2015). Total energy expenditure during floods (Costa and O'Connor, 1995) can be equally important as stream power and lateral confinement in estimating total sediment

transport (e.g., Wicherski et al., 2017). In contrast to fire studies, studies on the geomorphic impacts of extreme floods have usually focused on the erosional changes, even though short-duration, high-energy floods may cause substantial and long-lasting sediment deposition (e.g., Magilligan et al., 2015; Brogan et al., 2017).

New technologies, such as repeat airborne laser scanning (ALS), offer the potential to greatly improve our ability to quantify and analyse post-fire sediment storage and erosion over time and space (sensu Passalacqua et al., 2015). However, the

decimeter-scale uncertainty in detecting elevation change means that ALS differencing is most useful in stream channels and



valley bottoms where there is a greater likelihood of detectable elevation changes. As suggested in Moody et al. (2013), the goal is to relate the measured volumetric changes to key controls such as rainfall amounts and intensity, burn severity, and geomorphic characteristics, and use these to help predict the type and likelihood of downstream effects.

In June 2012 the High Park Fire (HPF) burned 350 km$^2$ of primarily montane forest just west of Fort Collins, Colorado, U.S.A. Within the HPF burn area we began intensively monitoring two similar ~15 km$^2$ watersheds to quantify post-wildfire geomorphic changes (viz., Brogan et al., in review). Subsequent rainfall-runoff floods created a unique comparison between the two watersheds, as one watershed was subjected to a very high intensity summer thunderstorm just one week after the fire was contained, and this caused very extensive downstream deposition that was not replicated in the other watershed. Fifteen months after burning an exceptionally large and long-duration mesoscale flood caused sustained high flows and channel erosion in both watersheds, and this severely altered the expected post-fire trajectory of persistent and progressively declining deposition. We were fortunate to have two ALS datasets to evaluate the post-fire changes prior to the mesoscale flood, and three ALS datasets to document the flood and subsequent post-fire effects over the following two years. This unique collection of sequential ALS data allows us to both quantify and compare the geomorphic changes due to the fire and the flood over time and space. We also can evaluate how the differences in the initial post-fire storms affected the relative effects of the mesoscale flood. The validity and our understanding of the ALS differences were greatly enhanced by other studies, including the intensive monitoring of 21 channel cross sections and longitudinal profiles in the two study watersheds (Brogan et al., in review), estimated peak flows due to the large convective storm one week after the fire was contained (Brogan et al., 2017, in review), the identification of rainfall thresholds for runoff and sediment delivery (Wilson et al., 2018), measured hillslope-scale erosion rates (Schmeer et al., 2018), and a more limited study of the hillslope erosion rates and channel changes in summer 2013 (Kampf et al., 2016). Together these data allow us to answer two key questions: 1) what are the spatial and temporal patterns of erosion and deposition following a wildfire and a large flood in small- to moderate-sized watersheds (0.1–15 km$^2$)? and 2) to what extent can these patterns be related to precipitation depths and intensities, burn severity, and valley and basin morphology.

## 2 Site Description

Two proximate and very similar watersheds, Skin Gulch (SG) and Hill Gulch (HG), were selected to investigate post-wildfire geomorphic changes (Figure 1). Both watersheds burned in the High Park fire, both drain north into the Cache la Poudre River, and they are similar in size at 15.3 and 14.2 km$^2$, respectively. Elevations range from 1890 to 2580 m in SG and HG is slightly farther east and lower at 1740 to 2380 m (Table 1). Average terrain slopes and drainage density for SG and HG are very similar at 23% and 24%, and 2.5 and 2.3 km km$^{-2}$, respectively. The two watersheds have nearly identical hypsometric curves with the bulk of the elevations falling within mid-elevations, with some flatter areas in the upper portions of each watershed. Land cover is primarily uninhabited wildland with 81% and 89% evergreen forest in SG and HG, respectively (Jin et al., 2013). SG is predominantly National Forest land, while HG is primarily privately owned. In each watershed there are several very small reservoirs that were presumably established as stock ponds. No control watershed could be identified due to the lack of sequential ALS data outside of the High Park fire.



Approximately 65% of each watershed was burned at moderate to high severity. In SG most of the area burned at moderate to high severity was in the upper headwaters, while in HG most of the moderate to high severity burned areas were in the lower portion of the watershed (Figure 1). Straw and wood mulch were applied in 2012 and 2013 to approximately 6% and 18% of the hillslopes in SG and HG, respectively. The underlying geology is primarily schist with scattered rock outcrops (Abbott, 1970, 1976; Braddock et al., 1988), and the soils are predominantly Redfeather sandy loams (HPF BAER Report, 2012; Soil Survey Staff, 2018). Headwater reaches range from wide shallow swales to steep and confined; the middle reaches generally are steep and confined with scattered floodplain pockets; and the downstream reaches are wider with mostly continuous floodplains. Sediment is stored predominantly in the channel bed and on the floodplains. The area is characterized as semiarid with mean annual precipitation of 450-550 mm (PRISM Climate Group, Oregon State University, http://prism.oregonstate.edu). Summer precipitation is usually derived from convective thunderstorms, while spring and fall storms tend to be lower intensity frontal storms. Approximately one-third of the annual precipitation falls as snow.

Streamflow in both watersheds was seasonal prior to burning, and the downstream mainstem channels were only about 1-2 m wide. After the fire streamflow noticeably increased and became perennial. One week after the fire had been contained a convective storm in SG generated large amounts of hillslope and upstream channel erosion, an estimated peak flow of nearly 30 $m^3$ $s^{-1}$ $km^{-2}$, and extensive downstream deposition (Brogan et al., 2017); this event is referred to as the 'convective flood' throughout the paper. No comparable storm occurred in HG, but in September 2013 a large mesoscale storm caused widespread and prolonged high flows in both watersheds. Peak flows were estimated to be 2.3–5.7 $m^3$ $s^{-1}$ $km^{-2}$ in SG and 0.9–1.4 $m^3$ $s^{-1}$ $km^{-2}$ in HG, with the range of values depending primarily on whether the peak flow is estimated using pre- or post-flood topography (Brogan et al., 2017, in review).

# 3 Methods

## 3.1 ALS preparation

In each of the four years after the fire an ALS dataset was collected over the entire burn area by the National Ecological Observatory Network (NEON) Airborne Observatory Platform. Each ALS dataset is referred to in this paper by the year and month of collection using the format of yyyymm, so the four NEON datasets are 201210, 201307, 201409, and 201506. A fifth ALS dataset, 201310, was collected by the U.S. Geological Survey (USGS) and Federal Emergency Management Agency (FEMA) in fall 2013 to help assess the damage caused by the September 2013 mesoscale flood. The four time periods between the five ALS datasets are referred to in this paper as T1, T2, T3 and T4. The 201307 ALS data in SG had substantial alignment issues, so we used OPALS (Orientation and Processing of Airborne Laser Scanning software Mandlburger et al., 2009) to improve the flightline alignment. Aerial photographs from 2008 were used to construct point clouds covering our study watersheds using structure-from-motion photogrammetry [unpublished data from S. Filippelli, Colorado State University, 2015]. Unfortunately these data did not allow for accurate volumetric differencing with respect to the first ALS dataset because extensive vegetation cover hampered the measurement of bare-earth elevations over most of the study area.





For each ALS dataset the raw point clouds were merged, ground classified, and clipped to our two study watersheds using LAStools (Isenburg, 2015). Ground classification parameters included: a buffer of 50 m; a step size of 5 m; and an extra fine search for initial ground points. From these processed point clouds we created digital elevation models (DEMs) with 1 x 1 m pixels (Isenburg, 2015). Care was taken to align all ALS DEMs as closely as possible using a Python script to calculate the

differences in slopes and aspects between each NEON DEM and the 201310 USGS/FEMA DEM (following the co-registration methodology from Nuth and Kääb, 2011). The resulting estimate of the XYZ translation required to rectify the location of each NEON DEM was repeated until translation changes in X, Y, and Z were less than 1 cm, or the required shift for that iteration was less than 2% of the overall required shift. Each point cloud was shifted by the computed translation, and DEM rasters were recreated from the translated point clouds. Finally, the rectified point clouds were compared to total station and RTK-GNSS

survey points to calculate the mean absolute error (MAE) as an indication of the accuracy of each ALS dataset.

### 3.2   Valley bottom and contributing area delineation

We used FluvialCorridor, an ArcGIS Toolbox that extracts a number of riverscape features (Roux et al., 2015), to delineate the valley bottoms in each watershed from the 201310 DEM. Defining a channel network is the first step, and for this we set a contributing area threshold of 0.1 km$^2$ based on local field surveys (Henkle et al., 2011). The valley bottom was then computed

and adjusted using a number of user-controlled input parameters, such as elevation threshold aggregation and disaggregation distances, buffer sizes, and smoothing tolerance. We adjusted these parameters until the valley bottom delineation satisfactorily matched aerial photographs and 2-m contour lines derived from the 201310 DEM.

Valley bottom polygons were segmented into 50-m long sections oriented in the downstream direction, yielding 595 segments in SG and 559 segments in HG. FluvialCorridor had difficulty characterizing valley bottoms for the headwaters of several

tributaries with gently sloping topography; unrealistically wide delineated valleys caused us to remove 89 and 56 segments in the headwaters of SG and HG, respectively. Another eight segments near the outlet of SG were excluded because the deposited sediment was repeatedly excavated by the state highway department (for example see Figure 10C in Kampf et al., 2016). Seven more segments in lower SG were excluded during T4 due to channel realignment and rehabilitation efforts, and one segment was excluded in lower HG during T4 due to the reconstruction of a house. A few other segments were removed from each

watershed due to small reservoirs and unreliable ground classification. Ultimately 490 segments in SG and 484 segments in HG were used for summarizing morphometrics (see section 3.4) and for statistical analysis (see section 3.7).

Contributing area polygons were delineated for each segment using a looped Python script that uses the 'Hydrology' toolset and 'Raster to Polygon' tool in ArcGIS. The resulting polygons were used to determine mean total rainfall and area-maximum maximum 30-minute rainfall intensity for each segment (see section 3.3 for more detail). Percent contributing area burned at

both high and moderate severity were determined for each segment using a burn severity ($BS$) map derived from RapidEye imagery and a multistage decision tree (Stone, 2015).



### 3.3 Precipitation

The amount and intensity of precipitation over the two study watersheds was determined from the National Weather Service WSR-88D Doppler radar in Cheyenne, WY, corrected with local daily rain gage data. We began by converting the dual-polarized one-hour precipitation accumulation (DAA) radar products into gridded precipitation estimates using a 0.5-km grid. The precipitation was summed for each grid cell from 0700 to 0700 local time to match the daily rain gage data. These radar estimates were then compared to the rain gage estimates to come up with a daily mean field bias (Wright et al., 2014):

$$B_i = \frac{\sum G_{ij}}{\sum R_{ij}} \tag{1}$$

where $B_i$ is the bias for day $i$, $G_{ij}$ is the daily rainfall for day $i$ and gage $j$, and $R_{ij}$ is the summed 24-hour rainfall for day $i$ and radar pixel containing $j$. Sources of gage data include four-inch diameter rain gages monitored by members of the Community Collaborative Rain, Hail & Snow (CoCoRaHS) Network (url: www.cocorahs.org), and tipping-bucket gages monitored by researchers at Colorado State University, the National Center for Atmospheric Research, and the U. S. Geological Survey. The number of rain gages used to compute the bias ranged from 36 to 97 depending on how many of the tipping-bucket gages were active and how many manual observations were recorded for a given day. These gages were located in and around our study watersheds, with the farthest gage being 40 km away.

Daily total rainfall and maximum 30-minute precipitation intensity ($MI_{30}$) were calculated from the bias-corrected DAA radar data for every 0.5-km grid cell across the HPF from October 2012 to November 2015. $MI_{30}$ was chosen over other intensity intervals (e.g., $MI_5$, $MI_{15}$, etc.) because it correlates best with peak flood discharge (Moody et al., 2013), and also is closely correlated with hillslope erosion rates from the HPF (Schmeer et al., 2018). Since volume changes over the intervals between ALS datasets represent cumulative geomorphologic effects, daily rainfall was summed for each of the four time periods. In contrast, the maximum $MI_{30}$ value between each ALS dataset was determined for each cell in each watershed. Finally, the mean total rainfall and the maximum $MI_{30}$ was computed for the upstream area of each channel segment for each DoD. This meant that the maximum $MI_{30}$ values for different cells within a given contributing area did not always originate from the same storm as the different summer thunderstorms were often very localized.

### 3.4 Topographic and hydraulic controls

A series of valley bottom, channel, and contributing area metrics, called morphometrics in this paper, were estimated for each 50-m segment. These data were correlated to the calculated volume changes to help determine possible controls on the volumes of erosion, deposition, and net change. A series of Python scripts were written to clip, extract and compute morphometrics directly from the DEMs and/or a combination of outputs from FluvialCorridor (e.g., stream network, segment polygons, valley widths). Stream networks for each ALS dataset were created for each watershed, and channel slope ($S$) for each segment was calculated using a linear regression on streamline elevations extracted from each ALS dataset at one-meter intervals. Topographic curvature ($\Delta S$) was quantified for each segment by calculating the slope of a linear regression where the channel slope of the segment and the two upstream segments were plotted against the distance upstream. A positive curvature



indicates a decrease in slope, while a negative curvature indicates an increase in slope. Valley width ($w_\mathrm{v}$) was computed at one-meter intervals along the valley centerline and an average width was calculated for each 50-m segment. Valley constriction and expansion ($\Delta w_\mathrm{v}$) was computed in the same way as $\Delta S$. Since the resolutions of the DEMs and aerial imagery were too coarse to accurately delineate the channels, channel width ($w_\mathrm{c}$) was estimated from a regional downstream hydraulic geometry

equation (Bieger et al., 2015):

$$w_c = 1.24 A^{0.435} \tag{2}$$

where $A$ is the drainage area in km$^2$ and channel width is in m.

    We defined channel confinement as the ratio of valley width to channel width ($C_r$). Unit stream power, a hydraulic control, is often a good predictor of erosion and deposition (e.g., Baker and Costa, 1987). Unit stream power is equal to:

$$\omega = \frac{\gamma Q S_f}{w_c} \tag{3}$$

where $\gamma$ is the specific weight of water (N m$^{-3}$), $Q$ is discharge (m$^3$ s$^{-1}$), and $S_f$ is the friction slope (m m$^{-1}$). Because continuous stage or flow data were not available, and given the potential uncertainty in the regression equation for $w_c$, we used the ratio of channel slope to valley width ($\frac{S}{w_v}$) as a proxy for stream power. Downstream changes in the slope-width ratio ($\Delta \frac{S}{w_v}$) were computed in the same way as $\Delta S$ and $\Delta w_\mathrm{v}$.

**3.5   Valley change**

DEMs of difference (DoDs) were computed using the geomorphic change detection (GCD) tool add-in for ArcGIS (gcd.joewheaton.org, version 6; Wheaton et al., 2010). GCD uses a fuzzy inference system (FIS) to propagate spatially explicit DEM uncertainties, and consequently the uncertainties in the DoD. Spatially propagated errors are much more accurate than assuming a uniform uncertainty, as the latter can lead to large errors in the calculated volumes of erosion and deposition (e.g., Wheaton et al., 2010;

Milan et al., 2011).

    Point quality, point density, and slope were included as membership functions in our FIS procedure. We assumed uniform point quality based on the accuracy of the ALS after adjustment (i.e., the MAE for each dataset). Point density was computed for each DEM pixel based on the point cloud, and slopes were derived directly from the DEM. After differencing the DEMs, pixels with elevation changes smaller than the spatially propagated errors were ignored, and the remaining values constitute

the thresholded DoD. The GCD tool also calculates total volumes of erosion, deposition, and net change, along with the uncertainty for each volume estimate. The uncertainties in the total volumes of erosion and deposition were computed by multiplying individual error heights times the pixel area and summing these. Uncertainty in each net volume difference was propagated from the corresponding uncertainties in erosion and deposition. Using the thresholded DoDs and our own Python script we computed the volumes of erosion, deposition, and net change for each 50-m segment for each time period.

The sign and overall magnitude of ALS-derived volumetric changes for the 50-m segments were compared to the surveyed changes at 10 cross sections in SG and 11 cross sections in HG (see Brogan et al., in review, for more information on the field data). The measured changes in cross-sectional area were multiplied by 50 m to obtain volumes that were then compared to the calculated ALS volume change for a given segment.



## 3.6 Removal of spurious vegetation artifacts

A visual check of the DoD results revealed the calculated volume changes were being affected by seasonal changes in leaf cover. For example, some locations had up to 3 m of deposition calculated from fall to summer (i.e., 201210—201307 (T1), 201409—201506 (T4)), and nearly identical amounts of erosion from summer to fall (i.e., 201307—201310 (T2)). Vegetation

5   issues were not immediately obvious in the 201310—201409 (T3) DoD, as both ALS datasets were collected in the fall. A raster-based algorithm was written to identify possible spurious changes due to changes in the deciduous leaf cover on a pixel-by-pixel basis for the DoDs that covered different seasons (i.e., T1, T2, and T4). An example of the algorithm's logic is as follows: If for a given pixel the change in both fall-to-summer differences (T1 and T4) were small, but the change from summer-to-fall (T2) was large compared to the T1 and T4 changes, it would indicate that vegetation was contaminating the signal at that location. This logic applies for other combinations of DoD differences, and takes the form of Algorithm 1.

---

**Algorithm 1** Vegetation removal algorithm

---

    **if** $DoD_{T1} - DoD_{T4} \leq \theta$ **and** $DoD_{T4} + DoD_{T2} \leq \theta$ **and** $DoD_{T2} + DoD_{T1} \leq \theta$ **then**

        pixel value = 0

    **else if** $DoD_{T1} - DoD_{T4} \leq \theta$ **and** $DoD_{T4} + DoD_{T2} \leq \theta$ **then**

        pixel value = 0

    **else if** $DoD_{T1} - DoD_{T4} \leq \theta$ **and** $DoD_{T2} + DoD_{T1} \leq \theta$ **then**

        pixel value = 0

    **else if** $DoD_{T4} + DoD_{T2} \leq \theta$ **and** $DoD_{T2} + DoD_{T1} \leq \theta$ **then**

        pixel value = 0

    **else**

        pixel value = 1

    **end if**

---

    In Algorithm 1, $DoD_{T\#}$ refers to the DoD for a given time period (i.e., T1, T2, or T4), and $\theta$ is a threshold in meters. We used this algorithm to classify each pixel as a 0 or 1, with 0 indicating a seasonal vegetation artifact when at least two of the three DoDs showed a difference in elevation change that was less than or equal to 1m ($\theta$). This raster of 1's and 0's was multiplied on a cell-by-cell basis for each DEM to exclude those pixels with a seasonal vegetation artifact for that DOD, and

15   the GCD tool was rerun to more accurately estimate the volume and uncertainty of geomorphic changes. Figure 2 shows an example of this vegetation filtering for a location in Skin Gulch that showed around 1 to 3 m of deposition from fall 2012 to summer 2013 (Figure 2A) and around 1 to 3 m of erosion from summer 2013 to fall 2013 before filtering out the seasonal artifacts (Figure 2B). A site visit in September 2016 verified the lack of such large vertical geomorphic change and confirmed a predominantly deciduous cover of narrowleaf cottonwood, Rocky Mountain maple, alders, chokecherry, and wild raspberries

20   (Figure 2C).





### 3.7 Statistical analysis of controls on erosion and deposition

Pearson correlation coefficients were calculated between the different site factors and the erosion, deposition and net volume changes in the 50-m segments for each of the four time periods and each watershed. The different site factors were total rainfall, $MI_{30}$, percent of contributing area burned at high and/or moderate severity, and drainage network morphometrics (as explained

in section 3.4). Since some of the morphometric variables changed from the beginning to the end of a given time period (i.e., $S$, $\Delta S$, $\frac{S}{w_v}$, and $\Delta \frac{S}{w_v}$), we calculated the correlations for each time period using both the before and the after values. We found negligible differences in the strength of the correlations depending on whether we used the before or after values, so we only present the results for the before values. Normalizing the net volume changes by contributing area generally did not improve the correlations, so these results also are not presented here. Correlations were also calculated after stratifying the data

by channel slope ($<$ or $\geq$ 4%) and contributing area ($<$ or $\geq$ 4 $km^2$), but these results are not presented. We did not stratify the data by physiographic unit or lateral confinement as suggested by Rinaldi et al. (2013) and Nardi and Rinaldi (2015) because the stream type in our two study watersheds is predominantly classified as cascade (Montgomery and Buffington, 1997). It should be noted that a positive correlation indicates increasing deposition or decreasing erosion with an increasing independent variable, while a negative correlation indicates decreasing deposition or increasing erosion. We recognize that

each stream segment is not necessarily spatially independent because upstream erosion or deposition can affect downstream segments or reaches, but auto-correlations of the dependent variables generally fall below $r = 0.5$ within five segments upstream or downstream. This initial correlation analysis provides a useful way to explore how morphologic and site characteristics are generally related to the magnitudes of erosion, deposition, and net volumetric change. In the results we primarily focus on correlation coefficients that are either greater than 0.32 and less than -0.32 (i.e., $R^2 > 0.10$).

## 4   Results

### Precipitation

Total rainfall and maximum 30-minute intensities varied considerably between each DoD time period, but the values were relatively similar within and between the two watersheds (Figure 3). The lowest amount of precipitation was in T1 with a mean of only 174 mm for SG and 185 mm for HG (Figure 3A). This period also generally had the lowest MI30 values other

than a few very localized high-intensity storms (Figure 3B). The second period included the large mesoscale storm and the rainfall from this storm was distributed relatively evenly across both watersheds (Kampf et al., 2016). Total rainfall over this three-month period ranged from 276 to 439 mm (Figure 3C), and this period tended to have the highest $MI_{30}$ values of 32-73 mm hr$^{-1}$ in SG and 36-106 mm hr$^{-1}$ in HG (Figure 3D). These higher values were due to convective summer thunderstorms as rainfall intensities during the mesoscale flood generally did not exceed 40 mm hr$^{-1}$ (Kampf et al., 2016).

The third period was nearly a year so it had relatively high total rainfall values but low $MI_{30}$ values (Figure 3F). As in T1 and T2, the variation in maximum $MI_{30}$ values was greater than the variation in total rainfall due to the high spatial variability of the summer thunderstorms. The total rainfall of about 260-450 mm during the fourth time period was less than T2 and T3





(Figure 3G). Mean $MI_{30}$ values also were lower at around 30 mm hr$^{-1}$ for SG and 38 mm hr$^{-1}$ for HG (Figure 3H), indicating less potential for hillslope erosion and downstream channel changes.

## 4.1 ALS data accuracy and valley morphometrics

Point density increased with each ALS dataset from a minimum of just under 1.2 pts/m$^2$ in the first ALS dataset to over 3.5
pts/m$^2$ for the last dataset in Skin Gulch and the next to last dataset in Hill Gulch. Mean absolute errors (MAE) of the final ALS point clouds in each watershed were only 9-13 cm, except for the MAEs of 23 and 15 cm for the first and second ALS datasets in HG, respectively.

The volume changes estimated from cross section data and the calculated volume changes from the ALS data for the corresponding segments generally fall along a 1:1 line except for one cross section for the second period in Skin Gulch and several
cross sections for the first time period in Hill Gulch (Fig. 4). The differences between these two datasets should not be too surprising given that the measured cross-section change was extrapolated to the entire 50-m segment. The main point is that the general agreement in the sign and magnitude of the ALS differencing and measured cross-section changes indicate that our ALS differencing is producing reasonable results.

The inherent comparability of SG and HG is further confirmed by the generally similar spatial distributions and trends
in channel slopes, valley widths, and confinement ratios. For the 490 segments in SG and 484 segments in HG used in our analyses 86% and 73% had channel slopes greater than 0.065 m m$^{-1}$, respectively, and were classified as cascade according to Montgomery and Buffington (1997). In SG and HG, respectively, 13% and 22% of the segments had channel slopes of 0.03 to 0.065 m m$^{-1}$, which would be classified as step-pool, and less than 2% and 5% of the segments had channel slopes less than 0.03 m m$^{-1}$, and would be classified as either pool-riffle or plane bed (Montgomery and Buffington, 1997). The few
channels with slopes less than 0.03 m/m are primarily in a few headwater areas, near tributaries, and towards the outlet of each watershed.

Valley widths tended to increase downstream, with the exception of certain headwater locations where FluvialCorridor had difficulty characterizing the valley bottoms. Approximately 80% of the valley widths in each watershed were between 10 and 40 m. As might be expected, confinement ratios tended to decrease downstream and were relatively similar in the two watersheds
with about 75% of the valley bottoms having values between 10 and 35, about 20% were greater than 35, and no segments had confinement ratios less than 5.

## 4.2 Spatial and temporal erosion and deposition volumes

T1 (201210–201307) included both spring snowmelt and some summer thunderstorms, and the DoD data show considerable variability in the spatial patterns of deposition and erosion within and between the two watersheds (Figures 5–9; see also Figures
A1–A4). In SG there was more deposition than erosion, which resulted in a net volume increase in the valley bottoms of nearly 8000 m$^3$ (Figure 5A). In the headwaters there was relatively little erosion or deposition, especially in the westward flowing channels in the easternmost part of the watershed (Figure 6A). In the middle portions of SG deposition was predominant (Figure 6A), and this was particularly evident along the main stem about 4-5 km above the outlet (Figure 8B). Lower in the watershed





there was net erosion and only limited deposition (Figures 6A and 8B). This erosion in the lower watershed was due primarily to snowmelt incising through the large amounts of sediment that had been deposited during the first summer after burning but before the first lidar dataset of October 2012. The greatest erosion of 130 m$^3$ in one 50-m segment was just downstream of a confluence about 2 km above the outlet (Figure 8B), which is where we observed showed tremendous deposition resulting from a large convective flood just one week after the fire (see reference to confluence and XS6 in Brogan et al., 2017, in review). At this location there is a very sharp decrease in channel slope and widening of the main valley (Figure 8A), which largely explains why there had been so much deposition. In general, however, the amounts of erosion or deposition were not obviously related to the morphometric characteristics in SG because the first ALS dataset in fall 2012 was only collected after the extensive hillslope erosion and downstream deposition in summer 2012.

In HG there was 19,000 m$^3$ of net deposition during T1, mostly in the main channels about 2-4 km above the watershed outlet (Figures 5B, 7A, and 9B). Much of this deposition was where the channel slopes decreased to less than $\sim$0.10 and valley widths increased to more than $\sim$30 m (Figure 9). Peak deposition of nearly 300 m$^3$ was in a segment about 2.5 km from the outlet, which is where the valley width abruptly increases to nearly 75 m and the slope drops below 0.05 (Figure 9). Similar to SG, the headwaters in HG had only minor erosion or deposition and there was a distinct lack of geomorphic changes in the westward-flowing channels in the easternmost portion of the watershed (Figure 7A). Aerial imagery and soils data (Soil Survey Staff, 2018) indicate that these areas are steeper with a greater density of exposed rock outcrops, suggesting shallower soils. These characteristics, combined with the steep narrow channels, limit the sediment supply as well as the capacity for deposition.

In September 2013, which was 15 months after the fire and during T2, the mesoscale flood caused widespread and often dramatic erosion in SG (Brogan et al., 2017, in review), with a total net erosion of 39,000 m$^3$. In SG erosion in the headwaters was minimal compared to the extensive channel changes in the middle and downstream reaches (Figures 6B, and 8C). In the middle reaches channel incision was common, especially in the narrower valley bottoms (see Figures 4.13D, 4.13E, and 4.13F in Brogan, 2018). Downstream channel widening and a few avulsions occurred where the valley was wide enough to contain a more continuous floodplain (see Figures 4.13B and 4.13C in Brogan, 2018). Many of the segments with the greatest erosion were in areas where there was simply more sediment to be eroded. These locations included floodplain pockets (e.g., $\sim$2.5 km, $\sim$2.9 km and $\sim$3.5 km from the outlet), tributary junctions (e.g., $\sim$1.4, $\sim$2.0 km and $\sim$3.7 km from the outlet), colluvial deposits from hollows (e.g., $\sim$1.8 km from the outlet), and deposition from a combination of processes (e.g., $\sim$0.6 km and $\sim$1.0 km from the outlet). The available sediment is believed to be a combination of pre-fire deposits accumulated over centuries to millennia (Cotrufo et al., 2016), while the extensive hillslope erosion in summers 2012 and 2013 added considerably more sediment that was readily accessible to the high flows during the mesoscale flood (Brogan et al., 2017, in review).

During T2 the greatest erosion in SG was at $\sim$1.8 km from the outlet where over 1,800 m$^3$ of sediment was removed (see Figures 4.13B and 4.13C in Brogan, 2018); the four segments upstream of this location also experienced substantial erosion, and similar to T1 the large amounts of erosion can be attributed to the very large amounts of deposition from the large convective flood that occurred just after the fire (Figures 6B and 8C; see also reference to confluence and XS6 in Brogan et al., 2017, in review). There also was up to 4.4 m of incision near a confluence in the middle of the SG. Overall the total erosion in SG





during T2 was 3.6 times larger than the total deposition during T1, and this discrepancy can be largely attributed to the fact that most of the post-fire sediment was deposited in summer 2012 before the first ALS survey. Similar to T1, there were little to no geomorphic changes in the westward-flowing channels in the easternmost part of each watershed (Figure 6B).

During T2 HG also experienced widespread erosion (Figures 6B and 7B; Brogan et al., 2017, in review), but the net volume

change was only two-thirds of the net volume change in SG (Figure 5). Similar to SG, the greatest erosion in HG occurred where there was more pre-fire sediment storage, including floodplain pockets (e.g., ~2.4 km, ~3.7 km and ~4.7 km), tributary junctions (e.g., ~2.2 km and ~3.3 km), and colluvial deposits from hollows (e.g., ~4.4 km; Figure 9). Substantial erosion also occurred where the hillsides constricted the valley width to less than 20 m; for example, there was over 800 $m^3$ and 1300 $m^3$ of erosion around 3.4-3.5 km and 3.8-4.0 km from the outlet (Figure 9C). Similar to T1, there were minimal geomorphic changes

in the westward-flowing channels in the easternmost part of the watershed (Figure 7B).

The pattern of erosion during T2 closely mirrored the depositional patterns from T1 (Figures 8 and 9), and this was particularly true for HG because there was much more deposition during T1 and qualitatively less deposition in summer 2012 prior to the first lidar dataset. For example, there was 2,300 $m^3$ of deposition in the valley bottom in HG between 2 and 3 km upstream of the outlet during T1, and this large amount of deposition was where the slope decreases to around 0.04 m m$^{-1}$ and the valley

width increases to 55 m. During T2 this same reach experienced 2,700 $m^3$ of erosion, or just slightly more than the amount of deposition during T1.

During T3 the patterns of erosion, deposition, and net change in both watersheds were similar in direction and location to T1 but smaller in magnitude (Figure 5). The magnitudes of change were more similar between the two watersheds in T3 than in T1 because there were no undocumented periods of erosion or deposition. In SG erosion in the southeastern headwaters

resulted in small alluvial fan deposits (Figure 6C), and again there also substantial deposition about 4-5 km from the outlet on the mainstem (Figure 8D). Farther downstream there was a more even balance between erosion and deposition (Figure 6C). The greatest erosion in SG occurred at a confluence around 3.7 km from the outlet where there was bank sloughing, which was largely a result of the channel incision and bank oversteepening that took place during the mesoscale flood in the previous time period (Figure 8D).

The T3 period in HG had more consistent deposition from the headwaters to the outlet than SG (Figure 7). The total volumes of erosion and deposition were slightly greater than in SG, but this difference was much smaller than the 2-3-fold difference measured in T1 (Figure 5). The largest depositional volumes of deposition were in the headwaters and the lowest portion of the watershed where the post-flood sediment was reworked and transported by spring snowmelt and summer thunderstorms (Figure 7C).

The magnitude of total net volume change in T4 was less than any of the other time periods (Figure 5). The overall pattern in both watersheds—like in T1 and T3—was deposition with very little erosion and a net volume change of just over 5,000 $m^3$. The similarity in total erosion, total deposition, and net volume changes between the two watersheds indicate a similarity in the primary driving processes of summer thunderstorms, hillslope erosion, downstream deposition, and erosion due to snowmelt. The absolute magnitude of these changes were the lowest in T4 as this was the third year after burning and the hillslopes





were recovering (Figure 5). As with the other time periods there generally were minimal changes in the headwaters of each watershed (Figures 6-9), and most of the larger volumetric changes were in the middle and lower portions of each watershed.

To summarize, the calculated volume changes for SG and HG were similar in their direction over the four time periods, and they also generally had roughly similar trends in magnitude (Figure 5). Net volume changes in T1, T3, and T4 for both channel networks were positive, and this plus other data show that the primary effect of the fire and subsequent rainstorms was erosion at the hillslope scale and deposition at scales larger than a few km$^2$. Over these three time periods both watersheds showed a decrease in the amount of geomorphic change over time, particularly in HG, as the estimated net volume change dropped from nearly 20,000 m$^3$ in the first period to just over 7,000 m$^3$ and 5,000 m$^3$ in the third and fourth periods, respectively (Figure 5B). In SG the net volumes over these same time periods also decreased from nearly 8,000 m$^3$ in T1 to over 6,000 m$^3$ and then 5,000 m$^3$ in T3 and T4, respectively (Figure 5A). Total deposition over all four time periods was just over 38,000 m$^3$ in SG and just over 46,000 m$^3$ in HG, while total erosion over all four time periods was similar with nearly 58,000 m$^3$ in SG and nearly 41,000 m$^3$ in HG. Seventy-eight percent and 72% of the total erosion in SG and HG, respectively, took place during T2 as a result of the September 2013 mesoscale flood. This means that in the absence of the highly unusual mesoscale flood the HPF would ultimately have caused extensive net deposition at scales greater than a few km$^2$.

## 4.3 Statistical analysis of controls on erosion and deposition

Pearson correlation coefficients indicate that several of the independent variables were highly correlated (Figure 10; see also Tables A1 and A2). These included: the percent area burned at high severity and percent area burned at moderate and high severity ($r = 0.99$ for both watersheds); slope-width ratio vs. channel width ($r = -0.59$ in SG and $r = -0.51$ in HG), contributing area versus channel width ($r = 0.94$ in SG and $r = 0.96$ in HG). As a result we removed percent area burned at high severity, slope-width ratio, and channel width from further analyses. We also removed the change in slope-width ratio and confinement ratio because of their dependency on other removed metrics.

Correlation coefficients ($r$) between the independent variables and the net volume change in each segment varied greatly between metrics and across time periods (Figure 10). The following sections summarize the key results for each time period in chronological order, and we report correlations are to indicate the direction as well as the magnitude of the relationship. Positive correlations indicate that increasing values of the independent variable were associated with either decreasing erosion or increasing deposition, while negative correlations indicate either increasing erosion or decreasing deposition.

In SG the absolute correlations ($|r|$) for net volume change during T1 never exceeded 0.17 (Figure 10), and this was primarily a result of the generally limited geomorphic change during this period (Figures 6A and 8B). The ALS data for T1 did not include the large amounts of deposition that we qualitatively observed in SG in the first three months after burning, but it did include the erosion of some of these deposits by subsequent spring runoff (Brogan et al., 2017, in review). Hence, the correlations were substantially greater when segment-scale erosion volumes were the dependent variable rather than deposition or net volume change (Figure 10). Segment-scale erosion was most strongly correlated with contributing area ($r = -0.56$), MI$_{30}$ ($r = -0.42$), and channel slope ($r = 0.33$). These results indicate that much of the erosion occurred in the lower gradient, wider downstream





reaches. We posit that the correlations for deposition and net volume change in SG would have been greater had the first ALS dataset captured the extensive post-fire deposition that we observed in the first summer after burning.

Overall the correlations in HG for T1 were slightly stronger in HG than in SG (Figure 10). In contrast to SG, deposition was more strongly correlated with the independent variables than either net volume change or total erosion. This difference is likely due to the greater magnitudes of deposition in the middle and lower reaches in HG relative to SG (Figure 7); the highest correlations with deposition were for contributing area ($r = 0.35$), $MI_{30}$ ($r = 0.34$), and $BS_m$ and $BS_{m+h}$ ($r = 0.33$ and -0.42, respectively).

Further investigation of the scatterplots indicate deposition predominated when contributing areas were less than about 4–5 $km^2$, while erosion dominated at contributing areas greater than about 4–5 $km^2$. Since the T1 period only included spring snowmelt and smaller convective storms in the first part of summer 2013, this indicates that the smaller convective thunderstorms had limited impact at larger scales, while elevated baseflow could cause significant channel changes if there was sufficient readily erodible sediment in the channels and valley bottoms.

Correlations for T2 were generally stronger than for any of the other three time periods, and this was primarily due to the substantial and consistent erosion resulting from the large mesoscale flood (Figure 10; Brogan et al., 2017, in review). In SG three metrics had $r > 0.32$ or $< -0.32$ with net volume change, and these included channel slope ($r = 0.35$), contributing area ($r = -0.63$), and $MI_{30}$ ($r = -0.36$). These results indicate increasing erosion in the downstream direction and nearly 40% of the variance in the amount of net change can be explained by $A$ alone. The correlations with erosion were generally stronger than the correlations with net volume change, and the highest correlation for any variable for any time period was between contributing area and erosion for T2 in SG ($r = -0.71$). Overall, the correlations with deposition as the dependent variable were weaker than the volumes of either net change or erosion (Figure 10).

As in SG, the correlations during T2 in HG were generally higher than for the other three time periods (Figure 10). The correlations for HG were not as high as for SG and this can be attributed to the lower volume changes in HG compared to SG (Figure 5). In HG two metrics had $r > 0.32$ or $< -0.32$ with net volume change, and these included channel slope ($r = 0.35$) and $MI_{30}$ ($r = -0.33$). Similar to SG, the correlations in HG generally improved when erosion was the dependent variable and decreased when deposition was the dependent variable (Figure 10).

Overall the volume changes in T2 were similar in magnitude but opposite in sign to the volume changes in T1 (Figures 8 and 9). Plots of the segment-scale net volume changes for T2 against the net volume changes for T1 show that much of the data plots along a line with a slope of -1 for SG and -0.8 for HG (Figure 11). This indicates that for many segments the volumes eroded primarily by the mesoscale flood were very similar to the volumes deposited in T1. However, the overall $R^2$ value was near zero in SG because a number of segments had far more erosion in T2 than was deposited in T1; these points plot well below the regression line and are shown in red in Figure 11A. A closer examination show that these segments are almost exclusively in the areas where there was tremendous deposition by the July 2012 convective storm and lesser deposition by other summer thunderstorms (Figure 11B) (Brogan et al., 2017) prior to the first ALS dataset. The shift in correlations from negative to positive, or vice versa, between T1 and T2 are particularly notable for channel slope ($r = -0.14$ in T1 and 0.35 in T2) and valley width ($r = 0.13$ in T1 and -0.17 in T2; Figure 10).



In HG the volumes of deposition in T1 and erosion in T2 were more similar (Figure 9) as indicated by the stronger $R^2$ value of 0.40, but again there is a cluster of points below the 1:-1 line (Figure 11C). The number and absolute magnitude of the differences between these points and the 1:-1 line is smaller than in SG, and this can be attributed to the smaller storms and associated deposition prior to the first ALS data set in October 2012. The segments below the 1:-1 line are almost exclusively in a major tributary draining an area burned at high severity (Figure 11D), and our field obserations indicate that this area also was subjected to deposition prior to the first ALS dataset (see Figure 3.9 in Brogan, 2018). Excluding these points from the regression increases the $R^2$ to 0.64, and this confirms the importance of the initial post-fire storms and the overall close relationship between the volumes of segment-scale deposition in T1 and the eroded volumes during T2. As in SG, many of the correlations in HG shifted from negative to positive, or vice versa, between T1 and T2, including channel slope ($r$ = -0.25 in T1 and 0.35 in T2), contributing area ($r$ = 0.28 in T1 and -0.24 in T2), and $MI_{30}$ ($r$ = 0.29 in T1 and -0.33 in T2; Figure 10).

In T3 and T4 the correlations between the independent variables and the segment-scale volume changes were generally low in both watersheds (Figure 10). The lower correlations can be attributed in part to the much lower amounts of erosion and deposition (Figure 5). The correlations generally had the same direction in T3 and T4 as T1, as each of these periods was primarily depositional. In SG the only correlations with net change with $r > 0.32$ or $< -0.32$ ($R^2 > 0.10$) were percent area burned at moderate severity ($r$ = -0.35) in T3 and total rainfall ($r$ = -0.33) in T4 (Figure 10). In HG none of the independent variables explained much more than 8% of the variation in net volume change, and the volumes of erosion and deposition again were only weakly correlated with the independent variables. For HG there were only three correlations with an $r > 0.32$ or $<$ -0.32, and these were for increasing segment-scale erosion with increasing contributing area ($r$ = -0.49) and valley width ($r$ = -0.38), and decreasing deposition with increasing percent area burned at moderate and high severity ($r$ = -0.38). The results for both watersheds indicate that the spring high flows continued to erode the relatively raw and enlarged channel created by the mesoscale flood.

## 5 Discussion

### 5.1 Mechanisms of watershed-scale post-fire erosion and deposition

As rainfall intensities exceed infiltration rates (e.g., Cammeraat, 2004; Kampf et al., 2016) the greatly enhanced hillslope runoff causes rapid expansion and incision of the headwater channels (Wohl, 2013). The increased runoff and increased connectivity transports the eroded sediment from the hillslopes down into the channel network (e.g., Prosser and Williams, 1998; Schmeer et al., 2018), with the finer particles being readily transported downstream as suspended load. In contrast, coarse sand and gravel are usually transported much shorter distances as bedload (e.g., Moody and Martin, 2001; Reneau et al., 2007), and are usually deposited in the wider, lower gradient reaches (e.g., Doehring, 1968; Anderson, 1976; Meyer et al., 1995; Moody and Martin, 2009).

The ash and sediment transported into the Cache la Poudre River after the High Park Fire greatly increased turbidities and suspended sediment concentrations (Writer et al., 2014), but these sediment inputs generally did not alter the channel morphology of the mainstem other than at a few tributary confluences, at a diversion dam, and further downstream where



the river emerged from the foothills into a wide valley bottom. Field data and observations both showed that fine sands, silts and clays did not comprise much of the post-fire deposits in either in the valley bottom of our two study watersheds or the mainstem of the Cache la Poudre River. This means that the topographic changes quantified by the ALS differencing primarily reflect the hillslope delivery, deposition, and some subsequent movement of the coarser bedload particles within our two study

watersheds.

The detailed, spatially-explicit calculations of erosion and deposition in our two study watersheds were only possible because of the relatively recent technology for differencing high-resolution topographic datasets. Our study was unique in terms of being able to compare five post-fire and post-flood ALS datasets, and the resulting maps of valley bottom changes show considerable spatial and temporal complexity that would not be possible from manual measurements (sensu Schumm, 1973).

Although complex, the DoDs clearly documented net overall deposition in both study watersheds during T1, T3, and T4, and net erosion in T2. This illustrates that–other than the mesoscale flood–the predominant post-fire effect is deposition in the channels and valley bottoms (Figure 5; see also Figure 3.21 in Brogan, 2018), and this preponderance of deposition over erosion is a typical post-fire response (e.g., Swanson, 1981; Morris and Moses, 1987; Moody and Martin, 2001; Wagenbrenner et al., 2006). Our surveyed cross sections and longitudinal profiles provide a more sensitive evaluation of post-fire changes (Brogan

et al., in review), but these only represent a small fraction of the channel network . In contrast, the ALS differencing covered the entire channel network, but the DoDs had much higher uncertainties due to alignment issues, horizontal displacement errors, interpolation errors, and errors associated with vegetation. Hence both types of data are needed to accurately and fully characterize the effects of the fire and subsequent flood, and they highlight the need to collect data at different spatial and temporal scales with different techniques and their associated levels of accuracy.

The smaller geomorphic changes in T3 and T4 are due to several factors. These include the ongoing hillslope vegetation recovery, reduction in headwater channel length (Wohl and Scott, 2017), and the relative paucity of large convective storms. In this study we also have to add another factor, which is the stripping and coarsening of the channel and valley bottoms due to the mesoscale flood. The poor accuracy of the first two ALS datasets in HG also means that the difference in the amount of deposition and net change between T1 and T3/T4 is almost certainly larger than what we calculated. These factors have resulted

in a sharp decline in hillslope runoff, erosion, and connectivity (Schmeer et al., 2018), and downstream channel geomorphic changes since September 2013. The presence and regrowth of riparian vegetation is another factor that can affect the amounts of erosion and deposition after fires and floods (e.g., Pettit and Naiman, 2007), and in SG there has been minimal riparian growth following the mesocale flood due to the very coarse substrate and depth to the water table. We argue that the stripping and coarsening of the channels and valley bottoms has resulted in a greatly reduced sensitivity to convective thunderstorms,

increased baseflows, and spring snowmelt (e.g., Brunsden and Thornes, 1979; Thomas, 2001; Phillips and Van Dyke, 2016; Rathburn et al., 2017; Fryirs, 2017; Brogan et al., in review).

Given that the uncertainty of our ALS differencing was usually 10-15 cm with a maximum of 23 cm, the ALS differencing was most able to detect elevation changes at tributary junctions and in larger channels and valley bottoms rather than on the hillslopes or in the smaller tributaries. Most of the largest volume changes were in downstream locations where channel

slopes were generally less than ~10% and valley widths were greater than ~30 m. The general trend of deposition at and





near confluences corroborates previous research (e.g., Benda et al., 2003; Nardi and Rinaldi, 2015), but in our case these changes were due to primarily to "standard" fluvial processes as there were few large debris flows after the High Park Fire and September 2013 mesoscale storm (Coe et al., 2014). The limited accuracy of the ALS differencing also leads us to posit that we underestimated deposition more than erosion because deposition tended to be more widespread and shallower compared to

the more localized and concentrated erosion.

## 5.2   Uncertainty, errors, and methodological issues in DEM differencing

It should be self-evident that future studies need to minimize the errors associated with DEM differencing if one is to accurately detect and quantify geomorphic changes, particularly in smaller streams. The challenges we faced working with the five different ALS datasets used in this study provides a series of useful insights into best practices for using repeat ALS data to

document geomorphic change after wildfires or other disturbances. First, ALS data collection must happen as soon as possible following the disturbance, particularly after fires as these landscapes are extremely sensitive to runoff, erosion, and channel change from even relatively small rainstorms (e.g., Shakesby and Doerr, 2006; Moody et al., 2013). Second, high-resolution topography should be repeated at the temporal resolution needed to distinguish and understand the seasonal effects of different driving forces (e.g., summer thunderstorms versus snowmelt). Recent advances in the use of drones rather than airplanes should

greatly facilitate more frequent lidar data collection (e.g., Tulldahl and Larsson, 2014), and allow researchers to be repeated at a sufficiently high temporal resolution to capture the effects of discrete storms and floods in addition to the seasonal changes characteristic of our study area. Drone-based structure-from-motion (SfM) photogrammetry offers an increasingly popular alternative to lidar and can result in much higher resolution data over time and space (e.g., Smith et al., 2016).

Third, repeat high-resolution topographic data often requires translational rectification to better match the different datasets.

In this study both vertical and horizontal translation was needed to more accurately match up the different the different ALS datasets, and thereby more accurately calculate elevation changes and associated volumes. Manual adjustments are laborious and non-repeatable, and our work was greatly facilitated by an automated approach to co-register the different point clouds Nuth and Kääb (2011). This approach, along with the availability of highly accurate RTK-GNSS field data (Brogan et al., in review), reduced the vertical uncertainties of most of our ALS data to 10-15 cm. Fourth, ALS data should be collected at low

altitudes with narrow flight pass widths, low scan angles, and good ground controls to improve the quality and density of the raw point clouds.

Automated GIS tools allow faster and easier characterization of the channel, adjacent topography and specific geomorphic features; examples include FluvialCorridor (e.g., Roux et al., 2015), River Bathymetry Toolkit (e.g., McKean et al., 2009), TerEx (Stout and Belmont, 2014), V-BET (Gilbert et al., 2016), and the Valley Confinement Algorithm (Nagel et al., 2014).

However, users must be aware of the limitations of these tools. FluvialCorridor provides objective valley bottom delineations that can be used over large spatial domains and facilitates longitudinal segmentation of the channel and valley bottom, but there were some problems in identifying valley margins when they was delineated near very steep slopes. In some cases the delineated valley bottom included the adjacent steep slope or rock outcrop, and the errors in estimating ground locations and elevations due to ALS interpolation error and horizontal displacement error (Hodgson and Bresnahan, 2004) near these steep





slopes can cause substantial errors in the DoD volume estimates (e.g., Heritage et al., 2009; Wheaton et al., 2010; Milan et al., 2011; Bangen et al., 2016). The inaccuracies in identifying the valley margins also caused higher elevation points to be included within a given segment, and these would bias the calculated valley bottom slopes. So our fifth cautionary point is that careful checking is needed of any automated process, and we found it necessary to sometimes manually delineate the valley bottoms, especially when the valley bottoms were directly abutted by steep terrain.

Sixth, techniques for computing elevation differences directly from point clouds are improving (e.g., Lague et al., 2013), but procedures to do so are still in their infancy (Passalacqua et al., 2015). In this study we initially tried to compute erosion and deposition volumes directly from the point clouds, but ultimately we used the standard DoD approach because there are lower uncertainties for raster-based methods when there are lower density point clouds (Hartzell et al., 2015). Another advantage of raster-based differencing is the mature suite of tools to calculate spatially-varying uncertainties, which improve the accuracy of volume change estimates compared to assuming a uniform uncertainty (e.g., Wheaton et al., 2010; Milan et al., 2011).

A key problem in this study was the large error due to the varying seasonal timing of the ALS datasets (i.e., leaf on versus leaf off). We developed an algorithm to remove unrealistically large elevation changes due to changes in canopy cover, and overall this reduced the mean calculated total erosion, total deposition, and net volume differences by 46% (s.d. = 16%), 54% (s.d. = 15%), and 22% (s.d. = 33%), respectively. However, the use of this algorithm increased the net volume change in T3 by 11% in SG and 25% in HG as it reduced the total deposition more than the total erosion. Careful checks of the DoDs and aerial imagery showed that this algorithm was still not able to always identify pixels with erroneous elevation changes due to changes in the vegetation heights between ALS datasets (e.g., Figure 2D). So our seventh point is that repeat ALS data should be collected at similar times of the year, preferably during leaf-off, to optimize the detection of elevation and volume changes. Again this shows that visual checks of DoDs are essential to detect a range of errors that otherwise are presumed to represent a real geomorphic change (e.g., Lane et al., 2004).

## 5.3 Controls on spatial and temporal patterns of geomorphic change

The linear regression results showed that post-fire volume changes were significantly correlated with rainfall depths and intensities, burn severity, and valley and basin morphology. However, none of the metrics had consistently strong coefficients of determination ($R^2$) with segment-scale volume changes over the different time periods and watersheds. We hypothesized that stronger correlations might be present when the watershed data were stratified by valley bottom slope or drainage area, but this did not greatly improve the strength or magnitude of the correlations. Alternatively, it has been suggested that better relationships could be attained by parsing the valley into more discrete geomorphic units (e.g., channel, floodplain, terrace) to reflect different dominant processes (e.g., Weber and Pasternack, 2017), but there was no easy way for us to do this across our study watersheds. Nevertheless, the correlations still provide useful insights into the controls on the volumetric changes. In particular, volume changes were consistently greater in segments with larger contributing areas, and where there were floodplain pockets, tributary junctions, and colluvial deposits. The largest volume changes were usually due to deposition in T1 and erosion in T2. Correlations in T3 and T4 generally were lower than in T1 and T2. This decrease in the strength of correlations is due in part to the lower magnitudes of erosion and deposition as the watersheds recovered from the fire, but also due to the reduced





sensitivity to channel change caused by the mesoscale flood. Overall the results strongly indicate that geomorphic changes in the channels and valley bottoms of our two study watersheds were largely controlled by sediment availability.

This assertion is further supported by the correlations between the segment-scale volumes deposited in T1 and the erosion volumes in T2, even though the T1 depositional volumes captured almost none of the large amounts of deposition that we observed in the first summer after burning (Figure 11; Brogan et al., 2017, in review). The importance of available sediment is also indicated by the scatterplots showing little net volume change in segments where the slope was much greater than about 0.2 m m$^{-1}$ (Figure 12). In these steep channels the bed is comprised of large, generally immobile sediment (e.g., Yager et al., 2012), and there is very limited potential to store the gravel and finer particles that represent most of the sediment eroded by surface runoff after a fire. To estimate potential geomorphic change in mountain catchments it may be more important to quantify where and how much sediment is available (e.g., Carling and Beven, 1989) rather than the spatial distribution of hydraulic and morphometric controls.

Areas of erosion and deposition are often highly correlated with the downstream gradient in stream power (e.g., Gartner et al., 2015; Yochum et al., 2017), but to our surprise none of our gradient metrics (i.e., $\Delta S$, $\Delta w_v$, $\Delta \frac{S}{w_v}$) were strongly correlated to net volume change, total erosion, or total deposition. Most of the largest volume changes occurred in segments where these gradients were close to zero, resulting in low correlation coefficients. These results again suggest that for our montane watersheds the spatial and temporal differences in sediment supply can better predict the volumes of erosion, deposition, and net change than local changes in slope or valley width.

Overall our correlations generally were not improved if erosion or deposition were used as the dependent variable instead of net volume change, but in some cases there were sharp differences in the correlations according to the dependent variable. For example, contributing area explained 50% of the total erosion in SG during T2 as well as 32% of the snowmelt erosion in T1. In each case contributing area led to a consistent increase in discharge along with the increase in sediment availability. Using deposition as the dependent variable generally did not improve correlations, with the primary exception being for T1 in HG as there was a stronger downstream trend in the initial volumes of post-fire deposition than in erosion or net change. In this study our efforts to quantify the controls on post-fire geomorphic changes were hindered by missing the first summer of deposition, the inability to detect smaller changes in elevation, and the residual vegetation effects.

Our understanding of the relationships between the independent variables and volume changes can be enhanced by more process-based research to couple estimated hillslope erosion rates to downstream volumetric changes. We found that valley morphometrics could explain some of the variations in post-fire deposition, erosion, and net volume change, and especially the erosion from the large mesoscale flood. Spatially explicit erosion models could be used to predict the spatial distribution of post-fire sediment inputs, and there is an urgent need to test the extent to which these varying sediment inputs are related to segment- and watershed-scale changes in sediment deposition and erosion. We might speculate that these predicted sediment inputs, which in this environment are most frequently caused by localized thunderstorms (e.g., Wagenbrenner and Robichaud, 2014; Kampf et al., 2016), may be more closely related to the observed volume changes than the valley and basin morphometric variables tested here. The observed correlations between volume changes, burn severity, and precipitation indicate that hillslope-scale erosion modeling could help improve our efforts to predict post-fire sediment storage and delivery as well as

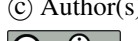



our understanding of the underlying processes. The goal would be a relatively complete sediment budget (sensu Vericat et al., 2017) that combines estimates of hillslope sediment delivery to the channel network (e.g., Schmeer et al., 2018) with spatially explicit estimates of volume changes over time in the downstream channels and valley bottoms.

The results from our two study watersheds show a clear commonality of controlling processes, but some substantial dif-

ferences in the magnitude of post-fire sediment storage and net volume change. The magnitude of post-fire effects on local residents and downstream water users depends on a suite of different factors, including the characteristics of the upstream burned area, the amount and intensity of precipitation, and the downstream watershed morphometry. After fires considerable funds are spent to reduce hillslope erosion risks (e.g., Robichaud et al., 2000), but there is a need to more rigorously evaluate the extent to which these hillslope risks are directly linked to the likelihood of a given downstream effect. Our research

helps identify where burned area emergency rehabilitation teams might focus post-fire rehabilitation efforts. Ecosystem and infrastructure concerns within or very near the burned area will require rehabilitation efforts immediately upstream. However, if the effects of greatest concern are much farther downstream, then post-fire treatments might best be focused on the tributary watersheds with relatively steep and narrow valleys that drain directly to the mainstem river and offer little potential for sediment storage. If ash and suspended sediment are of primary concern, rehabilitation efforts should focus on rapidly increasing

the amount of ground cover on the hillslopes as these materials, once introduced to the stream system, will be readily carried downstream. A more rigorous understanding of the controls on erosion and sediment storage, and the potential for longer-term storage of post-fire sediment, can help prioritize post-fire hillslope rehabilitation treatments and identify downstream locations with the greatest risk for post-fire sediment deposition.

## 6  Conclusions

Fires can induce tremendous amounts of overland flow and hillslope erosion, and these can cause profound erosion and deposition throughout the channel network. This study analyzed post-fire changes in the channels and valley bottoms in two 15 km$^2$ watersheds for three years after the 2012 High Park Fire. Field observations and a detailed analysis of channel and valley bottom changes from differencing five sequential airborne laser scanning datasets show the primary effect of the fire was deposition following summer thunderstorms with smaller amounts of incision from spring runoff. This sequence was interrupted

by a very unusual and large sustained flood in September 2013, 15 months after the fire, that eroded nearly all of the post-fire deposition along with much of the pre-fire valley bottom deposits. In the following two years there was much less deposition as the hillslopes recovered, and much less erosion as so much of the available sediment had been removed by the September 2013 mesoscale flood.

Precipitation depths and intensities, percent area burned at high and moderate severity, and valley and basin morphology

were weakly to moderately correlated with segment-scale volumes of deposition, erosion, and net change. This suggests that it is possible to identify areas with a greater potential for geomorphic change and hence a greater sensitivity. Our work shows that those areas with more deposition and sediment availability have the greatest potential for subsequent geomorphic change.



These locations include segments with lower slopes, tributary junctions, colluvial deposits and floodplain pockets, and wider valleys where there are more extensive and continuous floodplains.

Our experience in processing ALS datasets indicates the need to: 1) collect ALS data as soon as possible following a disturbance; 2) with a sufficient frequency to capture the effects of different driving forces; 3) at similar times of the year, preferably during leaf-off, to avoid vegetation artifacts; 4) establish good ground controls; 5) use an automated approach to co-register the point clouds; and 6) calculate spatially-varying uncertainties. Drones and structure-from-motion should greatly facilitate the collection of more frequent, high-resolution elevation data.

Future research should be aimed at investigating post-fire sediment routing from hillslopes through channel networks, quantifying geomorphic changes at shorter temporal scales, and evaluating how geomorphic changes vary among specific geomorphic units (e.g., channel, floodplain, pools, bars, etc.). Our ability to rigorously address these research needs is rapidly increasing as repeat high resolution topographic data become more readily available. Our results are an initial step towards more rigorously identifying downstream areas with higher sensitivity to geomorphic change, and thereby helping guide future post-fire mitigation efforts.

*Data availability.* Data associated with this manuscript can be accessed from the Colorado State University Digital Repository (Nelson and Brogan, 2019) (http://dx.doi.org/10.25675/10217/193080).

*Author contributions.* D.J.B. performed the analyses, collected field data, and wrote the manuscript. P.A.N. and L.H.M. assisted with the analysis and interpretation of data, writing, and editing of the paper.

*Competing interests.* No competing interests are present.

*Acknowledgements.* This work was supported financially by the National Science Foundation (EF-1250205, EF-1339928, and EAR-1419223), U.S. Department of Agriculture National Institute of Food and Agriculture Hatch project (1003276), the Arapaho-Roosevelt National Forest, and the USDA Forest Service National Stream and Aquatic Ecology Center. Airborne laser scanning was provided by the National Ecological Observatory Network, a project sponsored by the National Science Foundation. This material is based in part upon work supported by the National Science Foundation under Grant No. DBI-0752017.



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



**Table 1.** General watershed metrics for Skin Gulch and Hill Gulch.

| Metric | Skin Gulch | Hill Gulch |
|---|---|---|
| Contributing area (km$^2$) | 15.3 | 14.2 |
| Elevation range (m) | 1842-2683 | 1723-2397 |
| Relief (m) | 841 | 674 |
| Mean slope (%) | 23 | 24 |
| Total stream length (km) | 39 | 33 |
| Drainage density (km km$^{-2}$) | 2.5 | 2.3 |
| Elongation ratio | 0.53 | 0.44 |



**Table 2.** Point density and average mean absolute error (MAE) for each ALS dataset for Skin Gulch and Hill Gulch, respectively. MAE was determined by the elevation difference between total station and RTK-GNSS survey points and interpolated ALS points.

| ALS dataset | Skin Gulch | | Hill Gulch | |
|---|---|---|---|---|
| | Point density (pts/m$^2$) | MAE (cm) | Point density (pts/m$^2$) | MAE (cm) |
| 201210 | 1.16 | 12 | 1.18 | 23 |
| 201307 | 2.00 | 11 | 2.21 | 15 |
| 201310 | 3.01 | 11 | 2.78 | 9 |
| 201409 | 3.27 | 12 | 3.82 | 10 |
| 201506 | 3.67 | 13 | 2.21 | 13 |







**Figure 1.** Location and burn severity of the (A) High Park Fire (HPF) in the Colorado Front Range of the western U.S., and elevations of (B) Skin Gulch and (C) Hill Gulch. The black diamond to the east of Laramie in (A) is the location of the KCYS Doppler radar station in Cheyenne, WY. The thick blue lines in each watershed represent the reach used to present longitudinal results in Figures 8 and 9.





**Figure 2.** Seasonal changes in vegetation led to spurious deposition during fall to summer DoDs (A), and (B) spurious erosion in the summer to fall DoDs. The valley bottom in (A) and (B) includes several woody deciduous species along with some ponderosa pine (C). (D) shows the remaining change after using our raster-based algorithm to reduce the errors due to leaf out and leaf drop. Red circle in (C) identifies the upper half of a person standing in the understory, and the pink star in (D) represents the approximate location of the photo in (C).





**Figure 3.** Total rainfall (mm) and maximum 30-minute intensity (mm hr$^{-1}$) for the time periods between each successive DoD for: (A, B) 201210 to 201307; (C, D) 201307 to 201310; (E, F) 201310 to 201409; and (G, H) 201409 to 201506. Within each panel Skin Gulch is the watershed on the left and Hill Gulch is to the right.





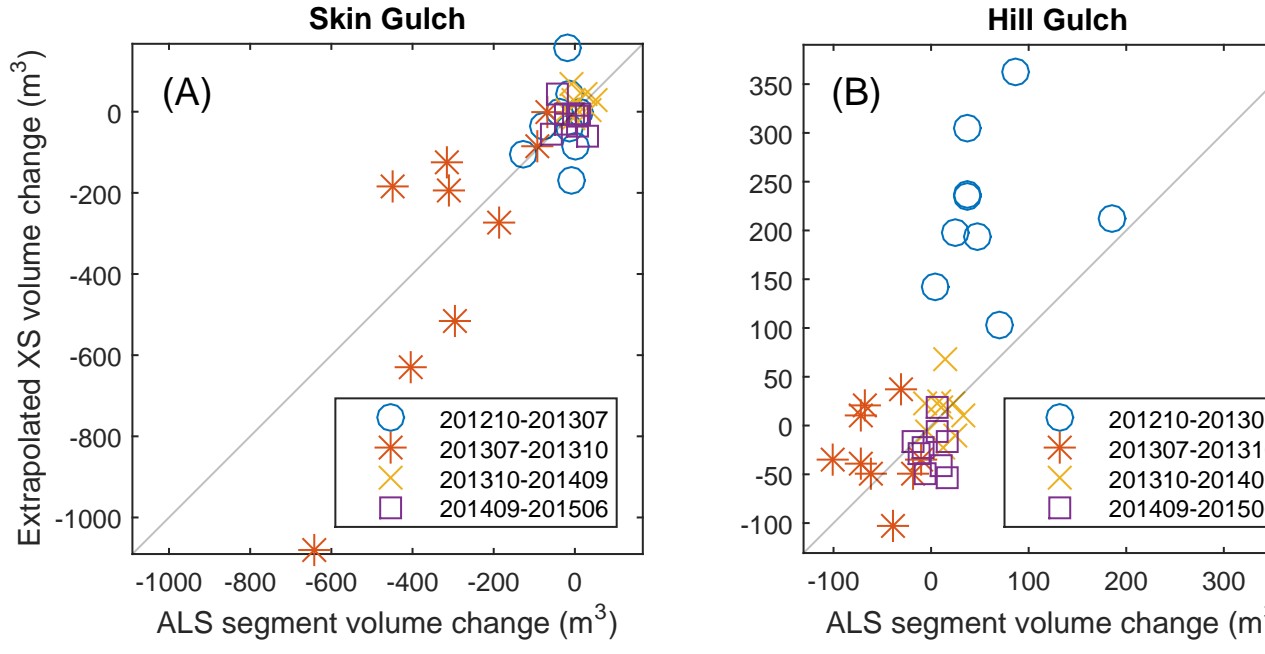

**Figure 4.** Comparison of the extrapolated cross section (XS) volume change and the ALS segment volume change for (A) Skin Gulch and (B) Hill Gulch. Diagonal lines are the 1:1 relationship.





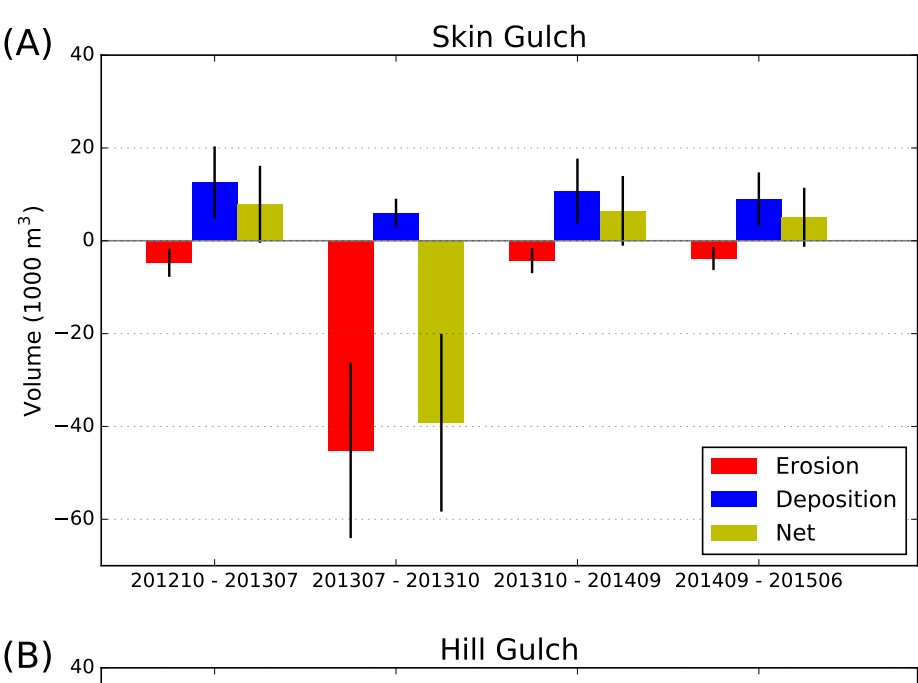

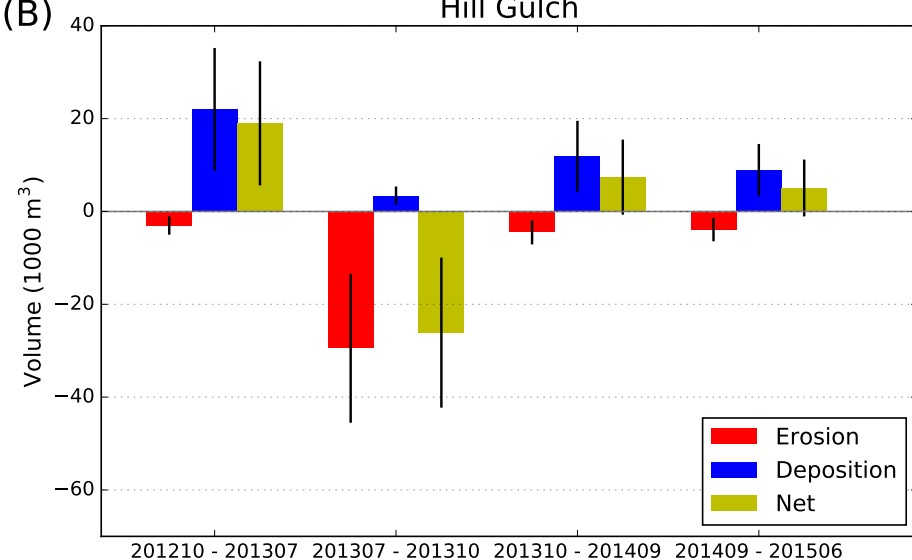

**Figure 5.** Total valley erosion, deposition, and net volume change for each time period for (A) Skin Gulch, and (B) Hill Gulch. Black vertical bars indicate the uncertainty in the volume estimates.

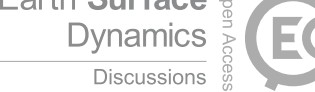



**Figure 6.** Net volume differences for each valley bottom segment in Skin Gulch for (A) 201210–201307, (B) 201307–201310, (C) 201310–201409, and (D) 201409–201506. Calculated volumes are not reported for the transparent segments due to unrealistically wide valley widths, repeat excavations, or the ground surface could not be reliably determined.







**Figure 7.** Net volume differences for each valley bottom segment in Hill Gulch for (A) 201210–201307, (B) 201307–201310, (C) 201310–201409, and (D) 201409–201506. Calculated volumes are not reported for the transparent segments due to unrealistically wide valley widths, repeat excavations, or the ground surface could not be reliably determined.



**Figure 8.** Longitudinal distributions in Skin Gulch of (A) elevation, channel slope, valley width and slope/width, and the corresponding change in volume for (B) 201210–201307, (C) 201307–201310, (D) 201310–201409, and (E) 201409–201506. Up and down arrows in (A) represent tributaries that enter the main channel from the right and left, respectively. Blue and red areas in (B)–(E) are deposition and erosion, respectively, and the black line is net volume change. Removal of excess sediment and restoration activities means that the data for the lowest 400 m were excluded for all time periods, and for the lower 700 m in (E).



**Figure 9.** Longitudinal distributions in Hill Gulch of (A) elevation, channel slope, valley width and flood power, and the corresponding change in volume for (B) 201210–201307, (C) 201307–201310, (D) 201310–201409, and (E) 201409–201506. Up and down arrows in (A) represent tributaries that enter the main channel from the right and left, respectively. Blue and red areas in (B)–(E) are deposition and erosion, respectively, and the black line is net volume change.





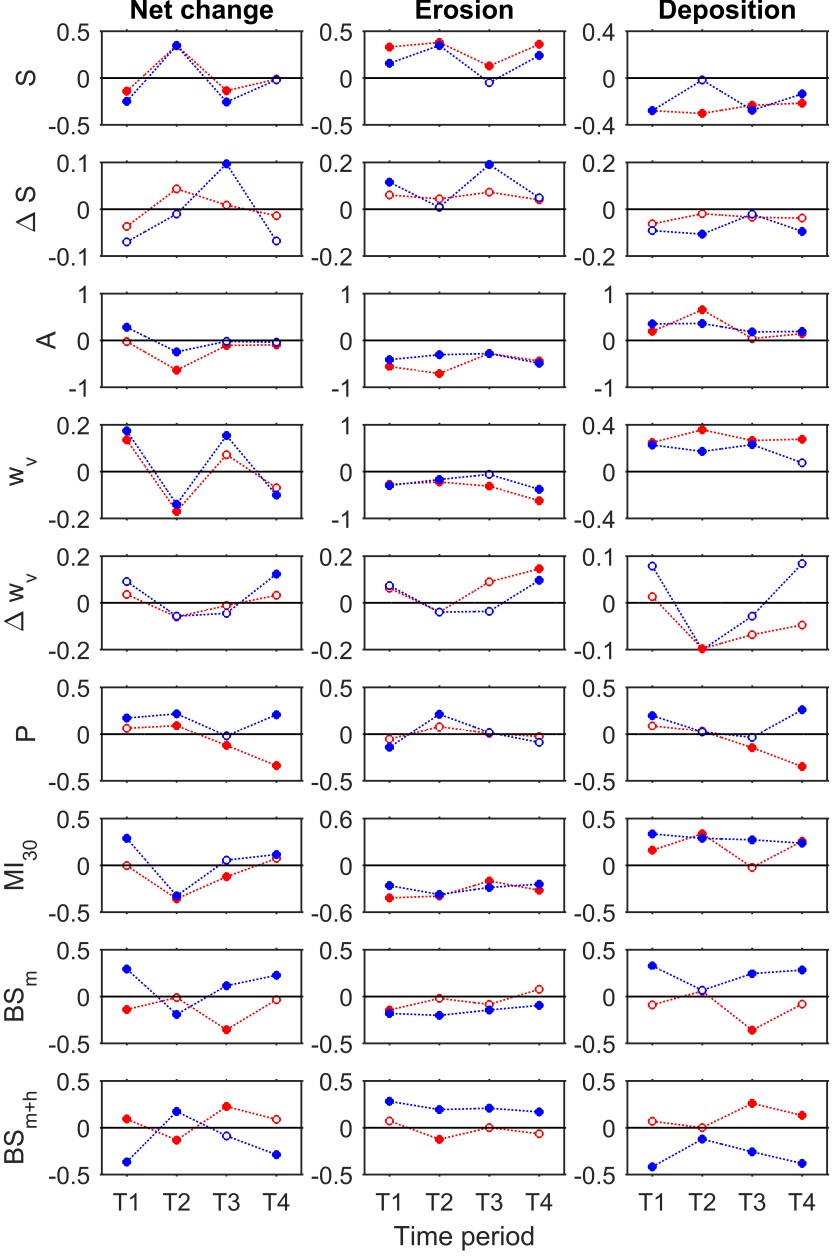

**Figure 10.** Correlation coefficients for Skin Gulch (red dashed lines) and Hill Gulch (blue dashed lines) for each time period between the independent metrics and the dependent variables of net volume change, total erosion, and total deposition. Time periods (T#) are for 201210–201307, 201307–201310, 201310–201409, and 201409–201506, respectively. Independent variables include channel slope ($S$), $\Delta S$, contributing area ($A$), valley width ($w_v$), change in valley width ($\Delta w_v$), total rainfall ($P$), maximum 30-minute intensity ($MI_{30}$), percent burned at moderate severity ($BS_m$), and percent burned at moderate-to-high severity ($BS_{m+h}$). Filled circles indicate significant correlations, p-value $\leq 0.05$.





**Figure 11.** Regression of the net volume change for each 50-m segment for T2 (the period including the large erosional mesoscale flood; 201307–201310) against the net volume change for T1 (the depositional period of 201210–201307) for (A) Skin Gulch and (D) Hill Gulch. The red x's in (A) and (D) are the segments with much more erosion in T2 than deposition in T1, causing them to deviate substantially from the dashed -1:1 line. The regression line and statistics for all of the data are shown in black, while the regression line and statistics in blue are for the truncated data after removing the red data points. (B) and (E) are burn severity maps of Skin Gulch and Hill Gulch, respectively, and the black boxes show the valley bottom segments in (C) and (F). The red segments in (C) and (F) are the red data points in (A) and (D).





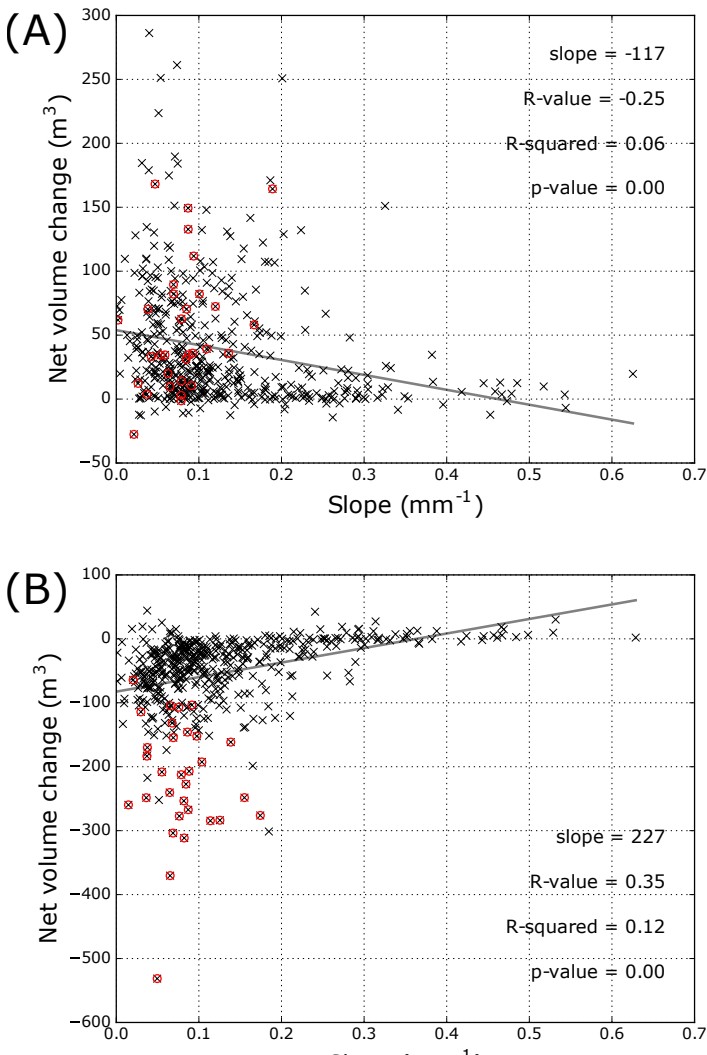

**Figure 12.** Scatterplot during of Hill Gulch for net volume change versus slope for (A) T1 and (B) T2. Red circles correspond to the segments highlighted in Figure 11.





**Table A1.** Pearson correlation coefficients ($r$) for the independent variables used in our statistical analysis in Skin Gulch. Independent variables include channel slope ($S$), $\Delta S$, contributing area ($A$), valley width ($w_v$), change in valley width ($\Delta w_v$), slope-width ratio ($\frac{S}{w_v}$), change in slope-width ratio ($\Delta \frac{S}{w_v}$), channel width ($w_c$), confinement ratio ($C_r$), total rainfall ($P$), maximum 30-minute intensity ($MI_{30}$), percent burned at moderate severity ($BS_m$), percent burned at high severity ($BS_h$), and percent burned at moderate-to-high severity ($BS_{m-h}$).

| $r$ | $S$ | $\Delta S$ | $A$ | $w_v$ | $\Delta w_v$ | $\frac{S}{w_v}$ | $\Delta \frac{S}{w_v}$ | $w_c$ | $C_r$ | $P$ | $MI_{30}$ | $BS_m$ | $BS_h$ | $BS_{m+h}$ |
|---|---|---|---|---|---|---|---|---|---|---|---|---|---|---|
| $S$ | - | | | | | | | | | | | | | |
| $\Delta S$ | 0.33 | - | | | | | | | | | | | | |
| $A$ | -0.54 | 0.03 | - | | | | | | | | | | | |
| $w_v$ | -0.48 | -0.08 | 0.37 | - | | | | | | | | | | |
| $\Delta w_v$ | 0.00 | -0.16 | -0.02 | 0.41 | - | | | | | | | | | |
| $\frac{S}{w_v}$ | 0.88 | 0.27 | -0.48 | -0.62 | -0.17 | - | | | | | | | | |
| $\Delta \frac{S}{w_v}$ | 0.15 | 0.68 | 0.05 | -0.20 | -0.52 | 0.32 | - | | | | | | | |
| $w_c$ | -0.65 | 0.02 | 0.94 | 0.42 | -0.04 | -0.59 | 0.06 | - | | | | | | |
| $C_r$ | 0.21 | -0.08 | -0.44 | 0.41 | 0.46 | -0.03 | -0.31 | -0.54 | - | | | | | |
| $P$ | 0.00 | -0.05 | 0.04 | 0.06 | 0.04 | 0.02 | -0.06 | 0.00 | 0.08 | - | | | | |
| $MI_{30}$ | -0.40 | 0.04 | 0.59 | 0.30 | -0.04 | -0.38 | 0.03 | 0.64 | -0.28 | 0.32 | - | | | |
| $BS_m$ | 0.16 | -0.07 | 0.05 | -0.14 | 0.07 | 0.24 | -0.10 | 0.01 | -0.11 | 0.17 | 0.00 | - | | |
| $BS_h$ | -0.16 | 0.07 | 0.02 | 0.13 | -0.10 | -0.23 | 0.10 | 0.08 | 0.02 | -0.08 | 0.14 | -0.84 | - | |
| $BS_{m-h}$ | -0.15 | 0.06 | 0.05 | 0.12 | -0.10 | -0.21 | 0.09 | 0.10 | -0.01 | -0.04 | 0.17 | -0.74 | 0.99 | - |





**Table A2.** Pearson correlation coefficients ($r$) for the independent variables used in our statistical analysis in Hill Gulch. Independent variables include channel slope ($S$), $\Delta S$, contributing area ($A$), valley width ($w_v$), change in valley width ($\Delta w_v$), slope-width ratio ($\frac{S}{w_v}$), change in slope-width ratio ($\Delta \frac{S}{w_v}$), channel width ($w_c$), confinement ratio ($C_r$), total rainfall ($P$), maximum 30-minute intensity ($MI_{30}$), percent burned at moderate severity ($BS_m$), percent burned at high severity ($BS_h$), and percent burned at moderate-to-high severity ($BS_{m+h}$).

| $r$ | $S$ | $\Delta S$ | $A$ | $w_v$ | $\Delta w_v$ | $\frac{S}{w_v}$ | $\Delta \frac{S}{w_v}$ | $w_c$ | $C_r$ | $P$ | $MI_{30}$ | $BS_m$ | $BS_h$ | $BS_{m+h}$ |
|---|---|---|---|---|---|---|---|---|---|---|---|---|---|---|
| $S$ | - | - | - | - | - | - | - | - | - | - | - | - | - | - |
| $\Delta S$ | 0.31 | - | - | - | - | - | - | - | - | - | - | - | - | - |
| $A$ | -0.45 | 0.01 | - | - | - | - | - | - | - | - | - | - | - | - |
| $w_v$ | -0.33 | -0.12 | 0.47 | - | - | - | - | - | - | - | - | - | - | - |
| $\Delta w_v$ | -0.05 | -0.21 | 0.03 | 0.45 | - | - | - | - | - | - | - | - | - | - |
| $\frac{S}{w_v}$ | 0.88 | 0.35 | -0.43 | -0.55 | -0.18 | - | - | - | - | - | - | - | - | - |
| $\Delta \frac{S}{w_v}$ | 0.18 | 0.80 | 0.03 | -0.20 | -0.44 | 0.36 | - | - | - | - | - | - | - | - |
| $w_c$ | -0.54 | 0.01 | 0.96 | 0.47 | 0.02 | -0.51 | 0.03 | - | - | - | - | - | - | - |
| $C_r$ | 0.34 | -0.12 | -0.41 | 0.37 | 0.33 | 0.06 | -0.24 | -0.53 | - | - | - | - | - | - |
| $P$ | 0.03 | 0.02 | -0.10 | -0.01 | -0.03 | 0.02 | 0.03 | -0.10 | 0.08 | - | - | - | - | - |
| $MI_{30}$ | -0.41 | 0.02 | 0.52 | 0.29 | 0.01 | -0.39 | 0.03 | 0.55 | -0.26 | 0.33 | - | - | - | - |
| $BS_m$ | -0.42 | 0.00 | 0.17 | 0.16 | 0.03 | -0.37 | 0.02 | 0.22 | -0.12 | 0.18 | 0.43 | - | - | - |
| $BS_h$ | 0.42 | -0.04 | -0.27 | -0.25 | 0.00 | 0.39 | -0.05 | -0.35 | 0.16 | -0.27 | -0.50 | -0.82 | - | - |
| $BS_{m+h}$ | 0.39 | -0.05 | -0.28 | -0.26 | 0.01 | 0.37 | -0.05 | -0.36 | 0.17 | -0.28 | -0.49 | -0.73 | 0.99 | - |





**Figure A1.** Total deposition for each valley bottom segment in Skin Gulch for (A) 201210–201307, (B) 201307–201310, (C) 201310–201409, and (D) 201409–201506. Calculated volumes are not reported for the transparent segments due to unrealistically wide valley widths, repeat excavations, or the ground surface could not be reliably determined.







**Figure A2.** Total erosion for each valley bottom segment in Skin Gulch for (A) 201210–201307, (B) 201307–201310, (C) 201310–201409, and (D) 201409–201506. Calculated volumes are not reported for the transparent segments due to unrealistically wide valley widths, repeat excavations, or the ground surface could not be reliably determined.



**Figure A3.** Total deposition for each valley bottom segment in Hill Gulch for (A) 201210–201307, (B) 201307–201310, (C) 201310–201409, and (D) 201409–201506. Calculated volumes are not reported for the transparent segments due to unrealistically wide valley widths, repeat excavations, or the ground surface could not be reliably determined.





**Figure A4.** Total erosion for each valley bottom segment in Hill Gulch for (A) 201210–201307, (B) 201307–201310, (C) 201310–201409, and (D) 201409–201506. Calculated volumes are not reported for the transparent segments due to unrealistically wide valley widths, repeat excavations, or the ground surface could not be reliably determined.