# Peer review of "Spatial and temporal patterns of sediment storage and erosion following a wildfire and extreme flood"

_Earth Surface Dynamics, 2018_

## Referee Comment (RC1) · Anonymous Referee #1 · 6 Mar 2019

General Comments: This paper investigates post-wildfire erosion using multitemporal lidar over time. I think some of the methods used by the authors are quite unique as compared to similar post-wildfire lidar studies (e.g. Pelletier and Orem, 2014 and Orem and Pelletier, 2015). In particular, I like their approach for removing DEM pixels that were determined to be disturbed by the canopy. That technique is novel, and I think it will provide a robust new method that will be embraced by the community. I also applaud the authors for their use of radar estimated rainfall and the approach used to correct it based on local rain gauge data. That allowed the authors to analyze spatially continuous rainfall data, which was useful for their overall analysis. I think that the general thrust of the paper is unique and I think that this manuscript is close to being

ready for publication. However, there are a few general suggestions that I will make, in addition to many specific suggestions below.

Consider re-writing section 4.2. Section 4.2 is a long chronological narrative, I understand the temptation to write it this way, because that is the way it unfolded in time. But it is really boring to read and fails to convey the salient points well. Consider organizing it in terms of drivers and response. This will help to generalize the paper beyond a case study.

Section 5.2 also needs some attention. The way that you break up the paragraphs is a little strange. I suggest abandoning the enumeration that you use "First, second, third." For example, on P17 line 19, why does that paragraph start with "third", and contain the "forth" point, but the next paragraph doesn't start with "Fifth"? I think the same points can be conveyed without this type of enumeration, and then paragraphs can be grouped by similar ideas.

I was also confused by your use of the term "sediment availability" in the discussion. At present, I don't see how your data speak to the sediment availability at all, and yet it is invoked as an explanation. I would consider either adding in data that relates to sediment availability, or rephrasing the sentences in which you point to sediment availability.

Lastly, I think it would be really helpful to synthesize these really unique results that moves beyond the case study. This could just be a paragraph in the discussion, but consider helping readers to see how the erosion/deposition sequence could be converted into something that might lead to more insight at different sites in the future.

This paper is really interesting, and despite my detailed comments, I enjoyed the approach, and I think that this manuscript will be a nice addition to the literature.

Specific Comments:

P2. L14: Rengers et al. modeled basin scale post-wildfire runoff. doi:

10.1002/2015WR018176

**ESurfD**

Interactive
comment

P3 L7-8: remove "very"

P3 L21: replace "small- to moderate-sized watersheds" to "stream channels" because it seems like all of your analysis is in the channels, not in the larger watershed.

P3 L28: I didn't see any hypsometric curves

P3 L30: Use a more specific term than "evergreen"

P4 L3: Say how straw and wood mulch were applied

P4 L6: add "channels" after the word combined

P5 L4: Are you going to post the python scripts anywhere?

P5 L18: Can you explain why you used the 50 m sections? You could have just analyzed the lidar on a pixel-by-pixel basis, so why create short reaches to analyze? This would benefit from some more explanation.

P5 L28-29: area-maximum maximum? Is that just a typo or does the second maximum go with the 30-min. rainfall intensity? Maybe rephrase so easier to understand.

P5 L30: Did you generate the burn severity map, if so mention that, if not say where it came from.

P6L5 is that 7am on one day and 7am the next day? Maybe make that more clear

P6L17: for the Moody 2013 ref, you should also ref. Kean et al. 2011 doi:10.1029/2011JF002005 See their figure 8 and reconcile that with your current statement .

P6L26: state goodness of fit for correlation

P6L31 ref a figure after "intervals"

P6 L31: I had (have) a really hard time visualizing exactly what you trying to say here

in the sentence that begins "Topographic curvature" Can you add a figure that is a schematic of what you are doing? A lot hinges on understanding this process, so I think it will be important that people don't miss what you are saying here.

P7L32: Why not just extract a line across the DoD at your X-S location?

P8L11: Cool approach!

P9L23: do you mean "The lowest TOTAL amount..."

P9L29: Does mesoscale refer to the 2013 flood? Make sure that is clear

P9L32: Add this rainfall to table 2

P10L4: D oyou have a way to estimate the size of the footprint of each laser point on the ground?

P10L9: I don't think it is accurate to say that "the ALS data ... generally fall along a 1:1 line". There seems to be a lot of deviation.

P10L15: reference tables after the word "ratios"

P10L18: Did you observe step pools?

P10L20: add "reaches" after channels

P10L31: add "within our LoD" after "deposition"

P11L10 :is the net deposition number (19000) from the ALS or your cross-sections? There is so much missing data that it is hard to believe this is a complete number. I am more interested in the longitudinal patterns than the specific volume estimates because of the missing data.

P12L26: There is so much missing data that it is hard to feel confident in the total volumes of erosion/deposition values in SG or HG. Consider focusing more on the patterns.

P13L5: "this plus other data . . ." what other data are you referring to?

P13L6: you mention hillslope scale, but I didn't think you had data on the hillslopes

P13L9: not quite a mass balance here, but I understand why

P13L16: "highly correlated" with what?

P13L33: "erosion occurred in the lower gradient" hmm that doesn't seem intuitive if slope is a major component of the driving shear stress. Can you help to explain why this makes sense somewhere?

P14L6: Is BS_m already defined? On page 5 BS is burn severity

P14L8: reference a figure after the word "scatterplots"

P16L1: What field data shows grain size?

P16L7: Add a ref like Passalaqua 2015 doi:10.1016/j.earscirev.2015.05.012 I also am not sure I agree with the word "recent" We're going on >20 years of lidar differencing

P16L11: "the predominant post-fire effect is deposition in the channels and valley bottoms" This is a more general statement than I think you are intending. For example, I don't think you would argue that this is necessarily true for the Poudre River. That is a channel/valley bottom, but it sounds like there was not extensive deposition there. So I suggest just refining the language to focus on the spatial scale at which you think it is representative.

P16L15 what fraction of the channel network does your ALS capture?

P16L28: Seems like you should mention the coarse substrate and depth to water table before this

P16L28: What exactly do you mean by stripping and coarsening of the channels?

P17L2: string "large" and add "documented" after "debris flows"

P17L15: I don't think you actually mean " allow researchers to be repeated" consider clarifying

P17L25: you qualitatively describe lidar here, why not just suggest a point density (pt/mˆ2) that you think would be good to shoot for.

P18L15: What proportion of the area was reduced from this approach?

P18L18: Your 7th point seems pretty obvious, but I guess the people at NEON didn't think about that. I thought it was typically standard practice.

P18L24: ref a figure at end of this sentence

P18L25: ref a figure at end of this sentence

P19L2: As far as I can tell, sediment availability is not something you measured (is it measureable?). Your results may allow you to make some inferences about sediment availability, but I don't think that the way things are presented right now allow you to say that the geomorphic changes were largely controlled by sediment availability.

P19L12-14: Maybe they aren't correlated because you calculated them across 50 m averages.

P19L16: What data do you have on sediment supply?

P19L21: Am I missing something? How do you know that sediment availability increased? What data are you pointing to for this statement?

P19L29-30: Spatially explicit models are being used: McGuire et al 2016 doi: 10.1002/2016JF003867; McGuire et al. 2017 doi: 10.1002/2017GL074243

P20L27: What makes sediment "available"? Figure 1: Mark the location of Laramie with a dot. What determines the thickness of the blue lines?

Figure 2: How did you calculate the maximum intensity?

Figure 10: Make sure to say these are "Pearson" correlation coefficients. They are

averaged for each time period, right?

Figure 11: Consider using equal axes in A and D.
* * *

---

## Referee Comment (RC2) · Anonymous Referee #2 · 18 Mar 2019

General: This manuscript reports on quantitative changes in erosion & deposition along 50 –meter length channel sections of two stream networks that experienced wildfire and flooding in a mountainous region of Colorado. Using DEMs of difference calculations from 4 time intervals spanning a total of ∼3 years, they show that significant volume changes in the 50-meter valley segments from erosion or deposition were correlated to contributing area, channel width, burn severity, channel slope, and rainfall intensity. The value of the manuscript is two-fold, because they develop thoughtful methods for analyzing the spatial and temporal pattern of sediment storage from repeat DEM data (including a canopy interference correction), and their conclusions about the landscape and meteorological controls on valley response can be used to predict

downstream risks in fire-prone landscapes. This is a very powerful paper with a nice dataset and is pretty close to being ready for publication.

While the authors were transparent in how they approached the study, there are some aspects that could be clarified simply to help the reader follow the rich dataset and somewhat involved analytical approach. Here are some suggestions that may help the presentation of the work:

-How did the authors land on 50-meter channel sections? Clearly this is a balance of resolving power and obtaining analytical units with meaningful change, but a few lines explaining the rationale of this length scale would be helpful

-Skin Gulch and Hill Gulch received significantly different volumes and intensities of precipitation over the study period: the magnitude of this difference should be generalized perhaps in a table (a row or two could be tacked on to Table 1) with maximum 30-minute rainfall rates measured over the time period or something that generalizes the total rainfall or intensity difference that the watersheds had. I appreciate the images in Figure 3 that show precipitation data in grids but I'm still left unclear on the magnitude of differences between the watersheds with regards to precipitation.

-I'm interested in the relationship between fire intensity and erosion/deposition measured in the channel sections. Fire intensity appeared to be one of the more significant predictors of net volume change in the channel, yet I'm unclear as to how and over what scale Burn Severity was calculated.

Brogan et al. find here that %burned at moderate to high intensity may be a good predictor of erosion/deposition measured in the channel; these results are consistent with the recent findings of Abrahams et al. 2018 (DOI: 10.1002/esp.4348) showing that burn severity was the biggest predictor of hillslope erosion in Fourmile Canyon, central Colorado.

Minor Comments:

The paragraph structure in several parts of the paper is weak, especially on pages 10-14: lots of small (2-4 sentence) paragraphs starting with the same word or phrase. Combine some of these short paragraph fragments into larger paragraphs that flow into one another.

On Figures 8 and 9, the general shape of the canyons is given in the upper pane (A. longitudinal profile, slope, valley width, etc.)- which DEM sources was used for these initial data? Because so many DEMS are used here, just be clear about which one is used for various visuals.

Figure 12: the x-axis title should be "channel slope".

---

## Author Comment (AC1) · 19 Apr 2019

**Response to Referee #1**

Please see our responses to each referee comment (in blue):

Anonymous referee #1: This paper investigates post-wildfire erosion using multitemporal lidar over time. I think some of the methods used by the authors are quite unique as compared to similar post-wildfire lidar studies (e.g. Pelletier and Orem, 2014 and Orem and Pelletier, 2015). In particular, I like their approach for removing DEM pixels that were determined to be disturbed by the canopy. That technique is novel, and I think it will provide a robust new method that will be embraced by the community. I also applaud the authors for their use of radar estimated rainfall and the approach used to correct it based on local rain gauge data. That allowed the authors to analyze spatially continuous rainfall data, which was useful for their overall analysis. I think that the general thrust of the paper is unique and I think that this manuscript is close to being ready for publication. However, there are a few general suggestions that I will make, in addition to many specific suggestions below.

We thank the referee for reviewing and provided general and specific suggestions. We have responded to each of your suggestions below. In most cases we agree to the suggested changes, and either have made or are making the changes in the manuscript.

Consider re-writing section 4.2. Section 4.2 is a long chronological narrative, I understand the temptation to write it this way, because that is the way it unfolded in time. But it is really boring to read and fails to convey the salient points well. Consider organizing it in terms of drivers and response. This will help to generalize the paper beyond a case study.

We appreciate this comment, and we (the three authors) had discussed how to best organize the paper. We understand the potential value of organizing the paper by processes ("drivers and response"), but this proved unwieldy and even more difficult to follow given the diverse responses over time, space, and between watersheds, and the different drivers in terms of convective storms, snowmelt, the large mesoscale flood, and the reduction in post-fire effects over time. We are revising Section 4.2 to eliminate some of the specific details and make it more succinct. This will lead to a greater focus on the key points and processes as suggested by the reviewer.

Section 5.2 also needs some attention. The way that you break up the paragraphs is a little strange. I suggest abandoning the enumeration that you use "First, second, third." For example, on P17 line 19, why does that paragraph start with "third", and contain the "forth" point, but the next paragraph doesn't start with "Fifth"? I think the same points can be conveyed without this type of enumeration, and then paragraphs can be grouped by similar ideas.

We will revise this section to either make the enumeration more consistent, or eliminate the enumeration altogether. We do feel that by listing the points it is easier for the reader to keep track of the larger organization and context with respect to the series of points that we are making.

I was also confused by your use of the term "sediment availability" in the discussion. At present, I don't see how your data speak to the sediment availability at all, and yet it is invoked as an explanation. I would consider either adding in data that relates to sediment availability, or rephrasing the sentences in which you point to sediment availability.

Our interpretation of "sediment availability" is based on: a) visual observation of sediment deposition in the valley bottoms and channels; and b) the estimates of deposition based on the lidar differencing. The underlying concept is that the post-fire sediment is much more readily erodible than the older sediment that is protected by vegetation. We will revise the manuscript to more clearly define what we mean by "sediment availability" and explain why the post-fire sediment was more accessible to erosion, especially during the mesoscale flood.

Lastly, I think it would be really helpful to synthesize these really unique results that moves beyond the case study. This could just be a paragraph in the discussion, but consider helping readers to see how the erosion/deposition sequence could be converted into something that might lead to more insight at different sites in the future.

Thanks for the suggestion. We will consider including text to better synthesize the results and use this to help expand the discussion.

This paper is really interesting, and despite my detailed comments, I enjoyed the approach, and I think that this manuscript will be a nice addition to the literature.

Thank you for this very positive overall summary of your evaluation of our manuscript. We have recently presented this story at several conferences, and have received a very positive response from the audience as this article does break some new ground in terms of comparing fires and floods, and also focusing on larger scale effects.

Specific Comments:
P2. L14: Rengers et al. modeled basin scale post-wildfire runoff. doi: 10.1002/2015WR018176

Thanks, we will add this citation.

P3 L7-8: remove "very"

We have deleted "very" as suggested by the reviewer.

P3 L21: replace "small- to moderate-sized watersheds" to "stream channels" because it seems like all of your analysis is in the channels, not in the larger watershed.

We agree, and have revised the manuscript to read "in the valley bottoms of small- to moderate-sized watersheds".

P3 L28: I didn't see any hypsometric curves

Yes, we did not include the hypsometric curves as we are only trying to indicate that the watersheds are very similar. The hypsometric curves are not critical to interpreting or understanding the results, and so we provided this fact but since we already have 12 figures we did not want to add yet another figure when this is not critical to our study. When describing a study area it is common to provide these kinds of descriptive statements without providing all the underlying data, and we believe that providing a statement about the hypsometric curves without providing the data is consistent with standard practices.

P3 L30: Use a more specific term than "evergreen"

We will add a sentence to indicate that the forest cover was primarily mostly ponderosa pine with some lodgepole pine and douglas fir at higher elevations and north-facing slopes.

P4 L3: Say how straw and wood mulch were applied

We have added "from helicopters" to clarify this. Wood chips were spread manually to a very small area (less than one hectare) that was mostly on a ridgetop and accessible by road.

P4 L6: add "channels" after the word combined

We presume that the reviewer means "confined" rather than "combined", and we have added the word "channels" to make this more explicit.

P5 L4: Are you going to post the python scripts anywhere?

The scripts referenced here piggyback off the FluvialCorridor ArcGIS add-on, and they were written specifically for our analysis so they would not be immediately useful to others. If a reader is interested they can contact us at the email address listed in the manuscript, and we will be happy to provide them with the scripts and any additional information, but they will have to make some modifications.

P5 L18: Can you explain why you used the 50 m sections? You could have just analyzed the lidar on a pixel-by-pixel basis, so why create short reaches to analyze? This would benefit from some more explanation.

The two main objectives of our paper were to: 1) characterize the spatially-explicit changes in sediment deposition, erosion, and net change over time throughout the channel network; and 2) relate these changes to both the morphometric characteristics and the characteristics of the contributing area (e.g., precipitation and burn severity). We therefore needed to do the analyses on larger segments rather than at the pixel scale as suggested. The segment length of 50 m is somewhat arbitrary, but we chose 50 m because: 1) this is an appropriate length to characterize the local morphometrics (i.e., channel slope and valley width) as well as the rate of change in these morphometrics given our valley bottom widths (i.e., long enough to minimize local noise but short enough to be relatively homogeneous); 2) the 50-m segment length matches up with the 50-m long longitudinal profiles that we were surveying at each cross-section; and 3) a rough rule of thumb is that longitudinal profiles should be about 10 times the channel width, and after the

2013 flood our channels ranged from about 2-10 m wide, so a 50-m long segment is "about right".

Given this comment and a similar comment from the second reviewer, we will insert one or two sentences in the text to explain and justify why we divided the channel network into 50-m segments.

P5L28-29: area-maximum maximum? Is that just atypo or does the second maximum go with the 30-min. rainfall intensity? Maybe rephrase so easier to understand.

This is not a typo, as we took the maximum value from the maximum values for all of the pixels. We have revised the wording to make this less confusing.

P5 L30: Did you generate the burn severity map, if so mention that, if not say where it came from.

At the end of the sentence we provide the reference for the burn severity map (Stone, 2015).

P6L5 is that 7am on one day and 7am the next day? Maybe make that more clear

On line 2 we state that the radar data were corrected with daily rain gage data, and on line 5 we state that the radar precipitation were summed from 0700 to 0700 to match the daily rain gage data. It also is standard observing practice that daily rainfall is measured from 0700 on one day to 0700 the following day, so we don't think that this needs any additional clarification.

P6L17: for the Moody 2013 ref, you should also ref. Kean et al. 2011 doi:10.1029/2011JF002005 See their figure 8 and reconcile that with your current statement.

It appears that Kean et al.'s (2011) Figure 8 indicates that I15 is most closely in phase with peak stage, but the lag between I30 and peak stage is only a few minutes. We can add a reference to Kean et al.'s paper here.

P6L26: state goodness of fit for correlation

We presume that by "goodness of fit values" the reviewer wants us to provide either the correlation coefficient ($r$) or the coefficient of determination ($r^2$). First, this would be a result, not in methods. Second, we cannot provide a goodness of fit values in the text because of the very large number of correlations that we compute for our results. We explicitly state on lines 25-26 that we collected a number of morphometric measurements (valley bottom, channel, contributing area), and on lines 26-27 we state that these data were correlated to the calculated volume changes. As shown in Table A1 and A2 we have 9 different morphometric characteristics and 5 different catchment characteristics, and when these are correlated to the volume changes in each watershed, we end up with 182 different correlations (91 for each watershed). The $r$ values from the analysis are displayed graphically in Figure 10. Hence it is not possible or appropriate to specify the correlation coefficients here.

P6L31 ref a figure after "intervals"

This section provides an explanation of how we calculated the channel slopes and changes in slope ("curvature"). We will improve the wording to better clarify how we calculated the slope and curvature, but a figure would be largely superfluous as the methodology is relatively simple and standard.

P6 L31: I had (have) a really hard time visualizing exactly what you trying to say here in the sentence that begins "Topographic curvature" Can you add a figure that is a schematic of what you are doing? A lot hinges on understanding this process, so I think it will be important that people don't miss what you are saying here.

As indicated in the previous comment, we are revising the text to make the methodology for calculating curvature more explicit. Curvature did not turn out to be a very important variable, so we would respectfully disagree that "A lot hinges on understanding this process."

P7L32: Why not just extract a line across the DoD at your X-S location?

That may be another approach to checking the validity of the lidar differencing. However, because the analyses in this paper focus on volume changes in the 50-m segments, we felt it was appropriate to compare the segment volumes to volumes calculated from field data.

P8L11: Cool approach!

Thank you!

P9L23: do you mean "The lowest TOTAL amount..."

We have revised this sentence to clarify that total precipitation was lowest during the T1 period.

P9L29: Does mesoscale refer to the 2013 flood? Make sure that is clear

We have defined the September 2013 flood as the "mesoscale flood" as this is consistent with other published accounts of this flood, and we have not used this term to refer to any other flood event. Inserting "2013" before "mesoscale flood" would imply that there was more than one mesoscale flood. We will closely examine every reference to the 2013 mesoscale flood to make sure that it is consistent, and that we explicitly note that there was only one mesoscale storm and flood.

P9L32: Add this rainfall to table 2

We are adding a short table to summarize the total rainfall and the maximum 30-minute intensities for each watershed and each time period.

P10L4: Do you have a way to estimate the size of the footprint of each laser point on the ground?

We do not have any information on the size of each laser footprint on the ground. Table 2 provides the point densities and the mean absolute error for each ALS dataset in each watershed, and we believe that this is sufficient to document the key point, namely that the data quality generally improved over time.

P10L9: I don't think it is accurate to say that "the ALS data ... generally fall along a 1:1 line". There seems to be a lot of deviation.

We agree that this wording was a bit strong, and we have revised this to note that the data generally plot close to the 1:1 line and then note the exceptions for the first time period in Hill Gulch and one cross section for the second period in Skin Gulch. We also note that one would not expect a perfect match because the cross sectional changes are extrapolated out to 50 m to obtain a volume.

P10L15: reference tables after the word "ratios"

We don't understand this comment, as the tables do not provide any data on the similarity in channel slopes, valley widths, or confinement ratios, and the purpose of this paragraph is to present a summary of these data to: 1) provide a more detailed description of these characteristics in the two watersheds; and 2) show that the two watersheds are relatively comparable with respect to their physiographic characteristics.

P10L18: Did you observe step pools?

Our field observations would suggest that there were a limited number of step-pool channels prior to the 2013 mesoscale flood, but these were generally smoothed out during the 2013 mesoscale flood. Since we have quantitative data on channel slope but only qualitative observations on channel type, we focus on channel slope. We also would argue that channel type is not an important control given the large-magnitude changes induced by the post-fire thunderstorms, snowmelt, and mesoscale flood.

P10L20: add "reaches" after channels

We have added "segments" to address the concern of the reviewer, as this terminology is consistent with our study.

P10L31: add "within our LoD" after "deposition"

By "LoD" we assume the reviewer is referring to our limits of detection. Since this caveat would apply to nearly every result pertaining to our DoD methodology, mentioning it here would imply that we should add it everywhere else. Since we are very explicit in noting that we can only evaluate elevation differences and hence volume changes in the channel and valley bottoms, we think it is best not to mention this caveat here to avoid the potential for confusion when reporting all our other DoD results.

P11L10 :is the net deposition number (19000) from the ALS or your cross-sections? There is so much missing data that it is hard to believe this is a complete number. I am more interested in the longitudinal patterns than the specific volume estimates because of the missing data.

Given the limited number of cross sections and our extended explanation of how we analysed the ALS data, we are confused that the reviewer could think that this volume was somehow derived from our cross section data. We assume by "so much missing data" the reviewer is referring to a net deposition calculation using field cross-sections; however, this number reflects ALS differencing, so we do not believe missing data is an issue here. Altogether our study does include 83% of the total channel length in Skin Gulch and 87% of the total channel length in Hill Gulch as stated in Section 4.1.
      Figures 6 and 7 present the complete data in space for each watershed and each time period, and the reader can use these to draw their own conclusions, and in Section 4.2 we tried to summarize the longitudinal patterns.

P12L26: There is so much missing data that it is hard to feel confident in the total volumes of erosion/deposition values in SG or HG. Consider focusing more on the patterns.

As noted in our previous comment, it is not clear to us what data are "missing". There are clear limitations on the change that we can detect, but this is true with every paper that attempts to calculate volume change from measured data. Figures 6 and 7 present the longitudinal data, and a close inspection of these figures show that the longitudinal patterns are very complex. A major result of our study is that the correlation analyses show that the volume changes cannot be explained to a high degree of certainty or resolution. We worked hard to try and identify clear and strong patterns, but our efforts had only limited success. Hence we see no way to focus more on the "patterns" as they are not nearly as clear as implied by the reviewer.

P13L5: "this plus other data..." what other data are you referring to?

We agree that this is ambiguous, and the inference was that "other data" was referring back to the list of previously published work on erosion and deposition after the High Park Fire (p. 3, lines 15-21 of our original manuscript). We will revise this to make it more clear that we are referring to other published studies on the High Park Fire.

P13L6: you mention hillslope scale, but I didn't think you had data on the hillslopes

Hillslope-scale erosion data were published previously, and we will revise this to explicitly indicate the source of the hillslope-scale erosion data and that the larger-scale deposition data are coming from the present paper and the recently published paper on our cross-section data (Brogan et al., *Geomorphology*, 2019).

P13L9: not quite a mass balance here, but I understand why

These are all net volume changes, and we will change "net volumes" to "net deposition" to make this more clear (per normal convention, positive values indicate net deposition and negative values indicate net erosion).

P13L16: "highly correlated" with what?

We appreciate the reviewer noting that this statement is ambiguous, and we will add words to note that the independent variables were correlated with volumes of erosion, deposition, and net change.

P13L33: "erosion occurred in the lower gradient" hmm that doesn't seem intuitive if slope is a major component of the driving shear stress. Can you help to explain why this makes sense somewhere?

Yes, this initially appears counterintuitive. The reason is that there was more post-fire deposition in the lower gradient downstream segments, and because there was much more sediment then there was more erosion. We have added the following wording at the end of this sentence to explain that the greater erosion was associated with greater amounts of post-fire deposition ("…and this is because these reaches generally had the greatest volumes of post-fire deposition and therefore had much more sediment that was readily available for erosion.")

P14L6: Is BS_m already defined? On page 5 BS is burn severity

We appreciate this comment. $BS_h$ and $BS_m$ refer to the proportion of the contributing area that was burned at high and moderate severity, respectively, and we have now defined these terms in methods.

P14L8: reference a figure after the word "scatterplots"

As noted above, the amount of deposition, erosion, and net volume change was correlated with each of the independent variables for each time period. Hence this result is a more general result, and in the interest of brevity we did not present any of the hundreds of scatterplots in the paper. A table of the overall correlations is included in the supplemental material.

P16L1: What field data shows grain size?

We will include a reference here to Brogan et al. (2019), where field grain size data are presented.

P16L7: Add a ref like Passalaqua 2015 doi:10.1016/j.earscirev.2015.05.012 I also am not sure I agree with the word "recent" We're going on >20 years of lidar differencing

Thank you for this, and we have revised the text to remove the word "recent" and made the text more explicit. In doing so we have obviated the need for a reference.

P16L11: "the predominant post-fire effect is deposition in the channels and valley bottoms" This is a more general statement than I think you are intending. For example, I don't think you would argue that this is necessarily true for the Poudre River. That is a channel/valley bottom, but it

sounds like there was not extensive deposition there. So I suggest just refining the language to focus on the spatial scale at which you think it is representative.

We appreciate this comment, but we do not say that deposition is universal, only that it is the predominant post-fire effect. We acknowledge appreciate the importance of spatial scale as suggested by the reviewer, and we therefore inserted the word "downstream" because incision is the predominant post-fire response at the hillslope scale. In confined valleys with large amounts of stream power there may not be widespread post-fire deposition until the channels and valley bottoms widen out, but even in the confined reaches of the Cache la Poudre River large amounts of coarse sediment were occasionally delivered into the river, and these did create relatively persistent alluvial fans. Hence the statement is more generally true, and we provide a series of references to support this statement.

P16L15 what fraction of the channel network does your ALS capture?

The point we were making is that the measured cross sections and longitudinal profiles represent only a small fraction of the channel network, so the DoD of repeated ALS surveys is needed in order to assess erosion and deposition over the entire channel network. This section has been rewritten to better contrast the higher temporal resolution field data with the ALS data, which have lower temporal resolution but can evaluate nearly the entire channel network.

P16L28: Seems like you should mention the coarse substrate and depth to water table before this

We agree that we should have mentioned the channel coarsening when reporting the effects of the mesoscale flood, even though the data on this came from the companion field paper, and we will add this to the results.
    We don't have any water table data, but infer this from the coarse substrate and the extensive channel incision that causes the stream water surface to be substantially lower than the elevation of the coarse deposits adjacent to the channel.

P16L28: What exactly do you mean by stripping and coarsening of the channels?

This statement is in reference to the changes induced by the mesoscale flood. As noted in the previous response, we will provide more description of the extensive channel erosion (stripping) and coarsening that occurred as a result of the mesoscale flood.

P17L2: string "large" and add "documented" after "debris flows"

Will do.

P17L15: I don't think you actually mean "allow researchers to be repeated" consider clarifying

Good catch, this will be changed to "allow researchers to collect data at a sufficiently…"

P17L25: you qualitatively describe lidar here, why not just suggest a point density (pt/m^2) that you think would be good to shoot for.

Good suggestion; we will change the text to note that the highest mean point density we had from our data was 3.8 pts m$^{-2}$. We will therefore recommend a minimum point density of 4 pts m$^{-2}$, noting that higher point densities would allow for a more detailed and accurate analysis.

P18L15: What proportion of the area was reduced from this approach?

Vegetation removal reduced the analyzed areas in both valleys by about 2%. We will point this out in the text, as it shows how significant an effect vegetation artifacts in just a small area can have on overall volume differencing calculations.

P18L18: Your 7th point seems pretty obvious, but I guess the people at NEON didn't think about that. I thought it was typically standard practice.

Yes, it should be obvious for volume differencing studies, although researchers with other interests (e.g., vegetation succession) may prefer data collection at different times of year.

P18L24: ref a figure at end of this sentence

We will refer to Fig. 10 here.

P18L25: ref a figure at end of this sentence

We will refer to Fig. 10 here.

P19L2: As far as I can tell, sediment availability is not something you measured (is it measureable?). Your results may allow you to make some inferences about sediment availability, but I don't think that the way things are presented right now allow you to say that the geomorphic changes were largely controlled by sediment availability.

Please see the response to the general comment above related to this topic. Again, we are interpreting sediment availability to observations or measurements of local deposition, which presumably provides sediment available for subsequent erosion and transport.

P19L12-14: Maybe they aren't correlated because you calculated them across 50 m averages.

This may be the case, but we think that 50 m is an appropriate window for computing valley widths, and should be a reasonable approximation of local slope in the absence of a sub-segment-scale knickpoint or other local discontinuity.

P19L16: What data do you have on sediment supply?

This sentence refers to sediment availability – we will change "supply" to "availability."

P19L21: Am I missing something? How do you know that sediment availability increased? What data are you pointing to for this statement?

This and other comments about sediment availability make it clear that the reviewer had a problem with our description of sediment availability. We will try to make the concept more clear in the revision.

P19L29-30: Spatially explicit models are being used: McGuire et al 2016 doi: 10.1002/2016JF003867; McGuire et al. 2017 doi: 10.1002/2017GL074243

Good point, we will cite these studies here.

P20L27: What makes sediment "available"?

Please see comments on this topic above – we will strive to clarify the concept in revision.

Figure 1: Mark the location of Laramie with a dot. What determines the thickness of the blue lines?

We will change the caption to not mention Laramie, and just say that the diamond represents the radar. The caption already designates the thickness of the blue lines.

Figure 2: How did you calculate the maximum intensity?

This is described in section 3.3 in the methods.

Figure 10: Make sure to say these are "Pearson" correlation coefficients. They are averaged for each time period, right?

Will do. Yes, these correlation coefficients represent relationships derived from ALS datasets collected at either end of the time period.

Figure 11: Consider using equal axes in A and D.

We prefer this figure as presented. Erosion and deposition rates in SG and HG differed enough that showing the plots at the same scale would compress the HG results and make the data more difficult to visualize.

---

## Author Response (AR1)

**Response to Referee #1**

**Please see our responses to each referee comment (in blue):**

Anonymous referee #1: This paper investigates post-wildfire erosion using multitemporal lidar over time. I think some of the methods used by the authors are quite unique as compared to similar post-wildfire lidar studies (e.g. Pelletier and Orem, 2014 and Orem and Pelletier, 2015). In particular, I like their approach for removing DEM pixels that were determined to be disturbed by the canopy. That technique is novel, and I think it will provide a robust new method that will be embraced by the community. I also applaud the authors for their use of radar estimated rainfall and the approach used to correct it based on local rain gauge data. That allowed the authors to analyze spatially continuous rainfall data, which was useful for their overall analysis. I think that the general thrust of the paper is unique and I think that this manuscript is close to being ready for publication. However, there are a few general suggestions that I will make, in addition to many specific suggestions below.

We thank the referee for reviewing and provided general and specific suggestions. We have responded to each of your suggestions below. In most cases we agree to the suggested changes, and the following documents each of our responses and the changes that we have made to the manuscript.

Consider re-writing section 4.2. Section 4.2 is a long chronological narrative, I understand the temptation to write it this way, because that is the way it unfolded in time. But it is really boring to read and fails to convey the salient points well. Consider organizing it in terms of drivers and response. This will help to generalize the paper beyond a case study.

We appreciate this comment, and we (the three authors) had discussed how to best organize the paper. We understand the potential value of organizing the paper by processes ("drivers and response"), but this proved unwieldy and even more difficult to follow given the diverse responses over time, space, and between watersheds, and the different drivers in terms of convective storms, snowmelt, the large mesoscale flood, and the reduction in post-fire effects over time. We have extensively revised Section 4.2 by deleting a number of the more specific details, combining some of the paragraphs, and emphasizing the key points and processes as suggested by the reviewer.

Section 5.2 also needs some attention. The way that you break up the paragraphs is a little strange. I suggest abandoning the enumeration that you use "First, second, third." For example, on P17 line 19, why does that paragraph start with "third", and contain the "forth" point, but the next paragraph doesn't start with "Fifth"? I think the same points can be conveyed without this type of enumeration, and then paragraphs can be grouped by similar ideas.

We appreciate the comment, and have revised this section to make the enumeration more consistent. We do feel that numbering the points makes it easier for the reader to keep track of the larger organization and the progression of the extended series of points that we are making based on our experience with sequential differencing of ALS datasets.

I was also confused by your use of the term "sediment availability" in the discussion. At present, I don't see how your data speak to the sediment availability at all, and yet it is invoked as an explanation. I would consider either adding in data that relates to sediment availability, or rephrasing the sentences in which you point to sediment availability.

Our interpretation of "sediment availability" is based on: a) visual observation of sediment deposition in the valley bottoms and channels; and b) the estimates of deposition based on the lidar differencing. The underlying concept is that the amount of erosion depends on the amount of available sediment, with the post-fire sediment being more readily erodible than the older sediment that is protected by vegetation. We have revised the manuscript to more clearly define "sediment availability" and show how sediment availability seems to be an important control on the amount of erosion in the channels and valley bottoms.

Lastly, I think it would be really helpful to synthesize these really unique results that moves beyond the case study. This could just be a paragraph in the discussion, but consider helping readers to see how the erosion/deposition sequence could be converted into something that might lead to more insight at different sites in the future.

Thanks for the suggestion. We have tried to streamline the last parts of the discussion to better emphasize how our results can have implications for predicting sediment deposition and erosion following disturbances.

This paper is really interesting, and despite my detailed comments, I enjoyed the approach, and I think that this manuscript will be a nice addition to the literature.

Thank you for this very positive overall summary of your evaluation of our manuscript. We have recently presented this story at several conferences, and have received a very positive response from the audience as this article does break some new ground in terms of comparing fires and floods, and also focusing on larger scale effects.

Specific Comments:

P2. L14: Rengers et al. modeled basin scale post-wildfire runoff. doi: 10.1002/2015WR018176

Thanks, we have modified the text and included this citation.

P3 L7-8: remove "very"

We have deleted "very" as suggested by the reviewer.

P3 L21: replace "small- to moderate-sized watersheds" to "stream channels" because it seems like all of your analysis is in the channels, not in the larger watershed.

We agree, and have revised the manuscript to read "in the valley bottoms of small- to moderatesized watersheds".

P3 L28: I didn't see any hypsometric curves

We did not include the hypsometric curves as we are only trying to indicate that the watersheds are very similar. The hypsometric curves are not critical to interpreting or understanding the results, and so we provided this fact. However, since the paper already has 12 figures we did not want to add yet another figure when this is not critical to our study. When describing a study area it is common to provide these kinds of descriptive statements without providing all the underlying data, and we believe that providing a statement about the hypsometric curves without providing the data is consistent with standard practices.

**P3 L30: Use a more specific term than "evergreen"**

We have added a sentence to provide more details on the forest cover.

**P4 L3: Say how straw and wood mulch were applied**

We have added "from helicopters" to clarify this. Wood chips were spread manually to a very small area (less than one hectare) that was mostly on a ridgetop and accessible by road.

**P4 L6: add "channels" after the word combined**

We presume that the reviewer means "confined" rather than "combined", and we have added the word "channels" to make this more explicit.

**P5 L4: Are you going to post the python scripts anywhere?**

The scripts referenced here piggyback off the FluvialCorridor ArcGIS add-on, and they were written specifically for our analysis so they would not be immediately useful to others. If a reader is interested they can contact us at the email address listed in the manuscript, and we will be happy to provide them with the scripts and any additional information, but they will have to make some modifications.

P5 L18: Can you explain why you used the 50 m sections? You could have just analyzed the lidar on a pixel-by-pixel basis, so why create short reaches to analyze? This would benefit from some more explanation.

The two main objectives of our paper were to: 1) characterize the spatially-explicit changes in sediment deposition, erosion, and net change over time throughout the channel network; and 2) relate these changes to both the morphometric characteristics and the characteristics of the contributing area (e.g., precipitation and burn severity). We therefore needed to do the analyses on larger segments rather than at the pixel scale as suggested. The segment length of 50 m is somewhat arbitrary, but we chose 50 m because: 1) this is an appropriate length to characterize the local morphometrics (i.e., channel slope and valley width) as well as the rate of change in these morphometrics given our valley bottom widths (i.e., long enough to minimize local noise but short enough to be relatively homogeneous); 2) the 50-m segment length matches up with the 50-m long longitudinal profiles that we were surveying at each cross-section; and 3) a rough rule of thumb is that longitudinal profiles should be about 10 times the channel width, and after the

2013 flood our channels ranged from about 2-10 m wide, so a 50-m long segment is "about right".

Given this comment and a similar comment from the second reviewer, we have added two sentences to explain and justify why we divided the channel network into 50-m segments.

P5L28-29: area-maximum maximum? Is that just a typo or does the second maximum go with the 30-min. rainfall intensity? Maybe rephrase so easier to understand.

This is not a typo, as we took the maximum value from the maximum values for all of the pixels. We have revised the wording to make this less confusing.

P5 L30: Did you generate the burn severity map, if so mention that, if not say where it came from.

At the end of the sentence we provide the reference for the burn severity map (Stone, 2015).

P6L5 is that 7am on one day and 7am the next day? Maybe make that more clear

On line 2 we state that the radar data were corrected with daily rain gage data, and on line 5 we state that the radar precipitation were summed from 0700 to 0700 to match the daily rain gage data. It also is standard observing practice that daily rainfall is measured from 0700 on one day to 0700 the following day, so we don't think that this needs any additional clarification.

P6L17: for the Moody 2013 ref, you should also ref. Kean et al. 2011 doi:10.1029/2011JF002005 See their figure 8 and reconcile that with your current statement.

It appears that Kean et al.'s (2011) Figure 8 indicates that I15 is most closely in phase with peak stage, but the lag between I30 and peak stage is only a few minutes. We have added a reference to Kean et al.'s paper here.

**P6L26: state goodness of fit for correlation**

We presume that by "goodness of fit values" the reviewer wants us to provide either the correlation coefficient (r) or the coefficient of determination ( $r^2$ ). First, this would be a result, not in methods. Second, we cannot provide goodness of fit values in the text because of the very large number of correlations that we compute for our results. We explicitly state on lines 25-26 that we collected a number of morphometric measurements (valley bottom, channel, contributing area), and on lines 26-27 we state that these data were correlated to the calculated volume changes. Figure 10 displays the correlations graphically for each of the nine independent variables and three dependent variables for each of the four time periods for each of the two watersheds (n=9 x 3 x 4 x 2, or a total of 216 correlations).

P6L31 ref a figure after "intervals"

This section provides an explanation of how we calculated the channel slopes and changes in slope ("curvature"). We have revised the text to better clarify how we calculated the slope and curvature, but a figure would be largely superfluous as the methodology is relatively simple.

P6 L31: I had (have) a really hard time visualizing exactly what you trying to say here in the sentence that begins "Topographic curvature" Can you add a figure that is a schematic of what you are doing? A lot hinges on understanding this process, so I think it will be important that people don't miss what you are saying here.

As indicated in the previous comment, we are revising the text to make the methodology for calculating curvature more explicit. Curvature did not turn out to be a very important variable, so we would respectfully disagree that "A lot hinges on understanding this process."

**P7L32: Why not just extract a line across the DoD at your X-S location?**

That may be another approach to checking the validity of the lidar differencing. However, because the analyses in this paper focus on volume changes in the 50-m segments, we felt it was appropriate to compare the segment volumes to the volumes calculated from an extrapolation of the field data.

**P8L11: Cool approach!**

Thank you!

P9L23: do you mean "The lowest TOTAL amount..."

We have revised this sentence to clarify that total precipitation was lowest during the T1 period.

**P9L29: Does mesoscale refer to the 2013 flood? Make sure that is clear**

We have defined the September 2013 flood as the "mesoscale flood" as this is consistent with other published accounts of this flood, and we have not used this term to refer to any other flood event. Inserting "2013" before "mesoscale flood" would imply that there was more than one mesoscale flood. We have closely checked the manuscript to ensure that we are consistent in how we reference the 2013 mesoscale flood, and that we explicitly note that there was only one mesoscale storm and corresponding long-duration flood.

**P9L32: Add this rainfall to table 2**

We have added a short table to summarize the total rainfall and the maximum 30-minute intensities for each watershed and each time period.

P10L4: Do you have a way to estimate the size of the footprint of each laser point on the ground?

We do not have any information on the size of each laser footprint on the ground. Table 2 provides the point densities and the mean absolute error for each ALS dataset in each watershed,

and we believe that this is sufficient to document the data quality and that the data quality generally improved over time.

P10L9: I don't think it is accurate to say that "the ALS data ... generally fall along a 1:1 line". There seems to be a lot of deviation.

We agree that this wording was a bit strong, and we have revised this to note that the data generally plot close to the 1:1 line and then note the exceptions for the first time period in Hill Gulch and one cross section for the second period in Skin Gulch. We also have adding text to explain that we would not expect a perfect match because the cross sectional changes are extrapolated out to 50 m to obtain a volume.

**P10L15: reference tables after the word "ratios"**

We don't understand this comment, as the tables do not provide any data on the similarity in channel slopes, valley widths, or confinement ratios. Similar to our earlier response, the purpose of this paragraph is to summarize these data to: 1) provide a more detailed description of these characteristics in the two watersheds; and 2) show that the two watersheds are relatively comparable with respect to their physiographic characteristics.

**P10L18: Did you observe step pools?**

Our field observations would suggest that there were a limited number of step-pool channels prior to the 2013 mesoscale flood, but these were generally smoothed out during the 2013 mesoscale flood. Since we have quantitative data on channel slope but only qualitative observations on channel type, we focus on channel slope. We also would argue that channel type is not an important control given the very large magnitude changes induced by the post-fire thunderstorms, snowmelt, and mesoscale flood.

**P10L20: add "reaches" after channels**

We have added "segments" to address the concern of the reviewer, as this terminology is consistent with our study.

**P10L31: add "within our LoD" after "deposition"**

By "LoD" we assume the reviewer is referring to our limits of detection. Since this caveat would apply to nearly every result pertaining to our DoD methodology, mentioning it here would imply that we should add it everywhere else. Since we are very explicit in noting that we can only evaluate elevation differences and hence volume changes in the channel and valley bottoms, we think it is best not to mention this caveat here to avoid the potential for confusion when reporting all our other DoD results.

P11L10 :is the net deposition number (19000) from the ALS or your cross-sections? There is so much missing data that it is hard to believe this is a complete number. I am more interested in the longitudinal patterns than the specific volume estimates because of the missing data.

Given the limited number of cross sections and our extended explanation of how we analysed the ALS data, we are confused that the reviewer could think that this volume was somehow derived from our cross section data. Nevertheless, we have revised the text to clarify how we estimated the changes in sediment volumes. We assume by "so much missing data" the reviewer is referring to a net deposition calculation using field cross-sections; however, this number reflects ALS differencing, so we do not believe missing data is an issue here. Altogether our study does include 83% of the total channel length in Skin Gulch and 87% of the total channel length in Hill Gulch as stated in Section 4.1.

Figures 6 and 7 present the complete data in space for each watershed and each time period, and the reader can use these to draw their own conclusions, and Figures 8 and 9 also show the longitudinal distribution of erosion and deposition for each time period in each watershed.

**P12L26: There is so much missing data that it is hard to feel confident in the total volumes of erosion/deposition values in SG or HG. Consider focusing more on the patterns.**

As noted in our previous comment, it is not clear to us what data are "missing". There are clear limitations on the minimum elevation change that we can detect, and we are very explicit about this (e.g., Table 1). Figures 6 and 7 present the longitudinal data, and a close inspection of these figures show that the longitudinal patterns are very complex. Our correlation analyses also show that the volume changes cannot be explained to a high degree of certainty or resolution. We worked hard to try and identify clear and strong patterns, but our efforts had only limited success. Hence we see no way to focus more on the "patterns" as they are not nearly as clear as implied by the reviewer.

**P13L5: "this plus other data..." what other data are you referring to?**

We agree that this is ambiguous, and the inference was that "other data" was referring back to the list of previously published work on erosion and deposition after the High Park Fire (p. 3, lines 15-21 of our original manuscript). We have revised the text to make it more explicit that we are referring to other published studies on the High Park Fire.

**P13L6: you mention hillslope scale, but I didn't think you had data on the hillslopes**

A paper on the hillslope-scale erosion results has already been published, and we have revised the text to more clearly indicate the source of the hillslope erosion data. We also have provided the reference for the deposition in the lower portions of HG and SG as measured from cross-sections and longitudinal profiles (Brogan et al., *Geomorphology*, 2019).

**P13L9: not quite a mass balance here, but I understand why**

These are all net volume changes, and per normal convention positive values indicate net deposition and negative values indicate net erosion.

**P13L16: "highly correlated" with what?**

We appreciate the reviewer noting that this statement is ambiguous, and have added wording to explain that we are first looking at the cross correlations among the different independent variables.

P13L33: "erosion occurred in the lower gradient" hmm that doesn't seem intuitive if slope is a major component of the driving shear stress. Can you help to explain why this makes sense somewhere?

Yes, this initially appears counterintuitive. The reason is that there was more post-fire deposition in the lower gradient downstream segments, and because there was much more available sediment then there was more erosion. We have revised the text to explain that the greater erosion was associated with greater amounts of post-fire deposition and that there was more sediment that could be readily eroded by snowmelt and lower intensity rainstorms.

**P14L6: Is BS\_m already defined? On page 5 BS is burn severity**

We appreciate this comment.  $BS_h$  and  $BS_m$  refer to the proportion of the contributing area that was burned at high and moderate severity, respectively, and we have now defined these terms in methods.

**P14L8: reference a figure after the word "scatterplots"**

As noted above, the amount of deposition, erosion, and net volume change was correlated with each of the independent variables for each time period. Hence this result is a more general result, and in the interest of brevity we did not present any of the 200+ scatterplots in the paper. Figure 10 does graphically present the correlation results, and a table of the overall correlations is included in the supplemental material.

**P16L1: What field data shows grain size?**

We will include a reference here to Brogan et al. (2019), where field grain size data are presented.

P16L7: Add a ref like Passalaqua 2015 doi:10.1016/j.earscirev.2015.05.012 I also am not sure I agree with the word "recent" We're going on >20 years of lidar differencing

Thank you for this, and we have extensively revised the text so that the section with "recent" is no longer present. The revisions also obviate the need for a reference.

P16L11: "the predominant post-fire effect is deposition in the channels and valley bottoms" This is a more general statement than I think you are intending. For example, I don't think you would argue that this is necessarily true for the Poudre River. That is a channel/valley bottom, but it sounds like there was not extensive deposition there. So I suggest just refining the language to focus on the spatial scale at which you think it is representative.

We appreciate this comment, but we do not say that deposition is universal, only that it is the predominant post-fire effect and we have provided extensive references and support for this statement in the introduction and results. We acknowledge appreciate the importance of spatial scale as suggested by the reviewer, and we therefore inserted the word "downstream" because incision is the predominant post-fire response at the hillslope scale. In confined valleys with large amounts of stream power there may not be widespread post-fire deposition until the channels and valley bottoms widen out. In the confined reaches of the Cache la Poudre River large amounts of coarse sediment were occasionally delivered into the river, and these did create relatively persistent alluvial fans. Hence the statement is more generally true, and we provide the material to support this statement.

**P16L15 what fraction of the channel network does your ALS capture?**

The point we were making is that the measured cross sections and longitudinal profiles represent only a small fraction of the channel network, so the DoD of repeated ALS surveys is needed in order to assess erosion and deposition over the entire channel network. This section has been rewritten to better contrast the higher temporal resolution field data with the ALS data, where the latter has lower temporal resolution but can evaluate nearly the entire channel network.

P16L28: Seems like you should mention the coarse substrate and depth to water table before this

We have deleted this sentence on the riparian vegetation as it was not crucial to the points we are making.

P16L28: What exactly do you mean by stripping and coarsening of the channels?

This statement is in reference to the changes induced by the mesoscale flood. As noted in the previous response, we will provide more description of the extensive channel erosion (stripping) and coarsening that occurred as a result of the mesoscale flood.

P17L2: string "large" and add "documented" after "debris flows"

We have deleted this sentence as it was not critical to our paper.

P17L15: I don't think you actually mean "allow researchers to be repeated" consider clarifying

Good catch, and we have revised this sentence.

P17L25: you qualitatively describe lidar here, why not just suggest a point density (pt/m2) that you think would be good to shoot for.

Good suggestion; we will change the text to note that the highest mean point density we had from our data was 3.8 pts m-2. We have added text recommending a minimum point density of 4 pts m-2, noting that higher point densities would allow for a more detailed and accurate analysis.

**P18L15: What proportion of the area was reduced from this approach?**

Vegetation removal reduced the analyzed areas in both valleys by about 2%. We will point this out in the text, as it shows how vegetation artifacts in just a small area can have a very large effect on the calculated differences in volumes.

P18L18: Your 7th point seems pretty obvious, but I guess the people at NEON didn't think about that. I thought it was typically standard practice.

Yes, it should be obvious for volume differencing studies, although researchers with other interests (e.g., vegetation succession) may prefer data collection at different times of year.

P18L24: ref a figure at end of this sentence

We have added a reference to Fig. 10.

P18L25: ref a figure at end of this sentence

Since we reference Figure 10 at the end of the previous sentence, we do not think it is necessary to repeat this reference here.

P19L2: As far as I can tell, sediment availability is not something you measured (is it measureable?). Your results may allow you to make some inferences about sediment availability, but I don't think that the way things are presented right now allow you to say that the geomorphic changes were largely controlled by sediment availability.

Please see the response to the general comment above related to this topic. Again, we are evaluating sediment availability on the basis of our field observations, field measurements at the cross sections and longitudinal profiles, and our DoD differencing. We present a number of lines of evidence to support our argument that sediment availability is a key control on the amount of subsequent erosion (e.g., Figure 11).

P19L12-14: Maybe they aren't correlated because you calculated them across 50 m averages.

This may be the case, but we think that 50 m is an appropriate window for computing valley widths, and should be a reasonable approximation of local slope in the absence of a sub-segment-scale knickpoint or other local discontinuity.

P19L16: What data do you have on sediment supply?

This sentence refers to sediment availability - we will change "supply" to "availability."

P19L21: Am I missing something? How do you know that sediment availability increased? What data are you pointing to for this statement?

This and other comments about sediment availability make it clear that the reviewer had a problem with our description of sediment availability. We have substantially revised the text to define sediment availability, describe how we assess sediment availability, and how sediment availability is related to the volume of subsequent erosion.

P19L29-30: Spatially explicit models are being used: McGuire et al 2016 doi: 10.1002/2016JF003867; McGuire et al. 2017 doi: 10.1002/2017GL074243

Good point, we have cited these studies here.

P20L27: What makes sediment "available"?

Please see our previous comments relating to the issue of sediment availability.

Figure 1: Mark the location of Laramie with a dot. What determines the thickness of the blue lines?

"Laramie" in the figure refers to Laramie County – county names are one of the layers on this map. The caption has been updated to reflect this. The caption already designates the thickness of the blue lines.

Figure 2: How did you calculate the maximum intensity?

This is described in section 3.3 in the methods.

Figure 10: Make sure to say these are "Pearson" correlation coefficients. They are averaged for each time period, right?

We have modified the caption to state that these are Pearson correlation coefficients. They represent the overall relationship between the change in volume for each time period versus the independent variable. So there is no "averaging", and we are confused by this comment.

Figure 11: Consider using equal axes in A and D.

We prefer this figure as presented. Erosion and deposition rates in SG and HG differed enough that showing the plots at the same scale would compress the HG results and make the data more difficult to visualize.

**Response to Referee #2**

**Please see our responses to each referee comment below (in blue):**

Anonymous referee #2: This manuscript reports on quantitative changes in erosion & deposition along 50-meter length channel sections of two stream networks that experienced wildfire and flooding in a mountainous region of Colorado. Using DEMs of difference calculations from 4 time intervals spanning a total of 3 years, they show that significant volume changes in the 50-meter valley segments from erosion or deposition were correlated to contributing area, channel width, burn severity, channel slope, and rainfall intensity. The value of the manuscript is two-fold, because they develop thoughtful methods for analyzing the spatial and temporal pattern of sediment storage from repeat DEM data (including a canopy interference correction), and their conclusions about the landscape and meteorological controls on valley response can be used to predict downstream risks in fire-prone landscapes. This is a very powerful paper with a nice dataset and is pretty close to being ready for publication.

**We greatly appreciate the referee's positive comments about the two-fold value of our paper, and that it is "pretty close to being ready for publication". ©**

While the authors were transparent in how they approached the study, there are some aspects that could be clarified simply to help the reader follow the rich dataset and somewhat involved analytical approach. Here are some suggestions that may help the presentation of the work:

-How did the authors land on 50-meter channel sections? Clearly this is a balance of resolving power and obtaining analytical units with meaningful change, but a few lines explaining the rationale of this length scale would be helpful

**As noted in our response to a similar comment from Reviewer 1, we have inserted two sentences in the text to explain why we divided the channel network into 50-m segments.**

-Skin Gulch and Hill Gulch received significantly different volumes and intensities of precipitation over the study period: the magnitude of this difference should be generalized perhaps in a table (a row or two could be tacked on to Table 1) with maximum 30-minute rainfall rates measured over the time period or something that generalizes the total rainfall or intensity difference that the watersheds had. I appreciate the images in Figure 3 that show precipitation data in grids but I'm still left unclear on the magnitude of differences between the watersheds with regards to precipitation.

**This again is something that Reviewer 1 noted, so we will add a short table to summarize the total rainfall and the maximum 30-minute intensities for each watershed and each time period.**

-I'm interested in the relationship between fire intensity and erosion/deposition measured in the channel sections. Fire intensity appeared to be one of the more significant predictors of net volume change in the channel, yet I'm unclear as to how and over what scale Burn Severity was calculated.

We first note that fire intensity is heat lost per unit time per unit flame length, while severity is the effect on the vegetation ("vegetation burn severity") and soils ("soil burn severity"). We presume that the reviewer is concerned with burn severity, as there are no data on fire intensity. In the methods we specify that we did have a burn severity map and provide a reference for this. We also state in Section 3.2 that for each segment we determined the percent of the contributing area that was burned at high and moderate severity, respectively (p. 5, lines 29-31 in the original manuscript). In response to this comment we are altering the wording in this sentence to make it more explicit: "Percent area burned at both high and moderate severity (BS) map ...(Stone, 2015)." (changes in bold).

Brogan et al. find here that %burned at moderate to high intensity may be a good predictor of erosion/deposition measured in the channel; these results are consistent with the recent findings of Abrahams et al. 2018 (DOI:10.1002/esp.4348) showing that burn severity was the biggest predictor of hillslope erosion in Fourmile Canyon, central Colorado.

The fact is that researchers have long recognized the importance of burn severity for predicting hillslope runoff and erosion, and we have already referenced some of the most directly relevant papers (e.g., Benavides-Solorio and MacDonald, 2001; Wagenbrenner et al., 2006). The problem is that burn severity is a categorical variable, so it is generally better to relate erosion rates to percent bare soil, as percent bare soil is a continuous variable that can be plotted directly against erosion rates, which is another continuous variable. We appreciate this new reference, and now refer to it in the discussion.

**Minor Comments:**

The paragraph structure in several parts of the paper is weak, especially on pages 10-14: lots of small (2-4 sentence) paragraphs starting with the same word or phrase. Combine some of these short paragraph fragments into larger paragraphs that flow into one another.

Yes, there are a lot of short paragraphs. We separated the paragraphs in order to make it more clear that we were switching topics or locations. We have extensively revised Section 4.2 in response to the first reviewer, and as we delete some of the details and focus on the broader story we have consolidated many of the short paragraphs that were a concern for the reviewer.

On Figures 8 and 9, the general shape of the canyons is given in the upper pane (A. longitudinal profile, slope, valley width, etc.)- which DEM sources was used for these initial data? Because so many DEMS are used here, just be clear about which one is used for various visuals.

We have added a sentence to the captions to make it clear that the data in the first panel are derived from the October 2013 DEM developed by the USGS.

Figure 12: the x-axis title should be "channel slope".

We appreciate this comment, and have changed both the x-axis labels and the caption so that slope is now explicitly labeled as "channel slope".

[revised manuscript text omitted]
 | M                                       | Skin Gul                 | ch                                                          | Hill Gulch               |                                                        |  |  |  |
|-------------|-----------------------------------------|--------------------------|-------------------------------------------------------------|--------------------------|--------------------------------------------------------|--|--|--|
|             | Months                                  | Total precipitation (mm) | $\underbrace{MI_{30} \ (\mathrm{mm} \ \mathrm{h}^{-1})}_{}$ | Total precipitation (mm) | $\underbrace{MI_{30} (\mathrm{mm}\mathrm{h}^{-1})}_{}$ |  |  |  |
|             | .8~ | 174 (156-234)            | 24 (11-85)                                                  | 185 (175-205)            | 17 (13-32)                                             |  |  |  |
|      | 3                                       | 366 (276–439)            | 49 (32-73)                                                  | 327 (302–439)            | 49 (36–106)                                            |  |  |  |
|      | 11                                      | 527 (441-634)            | 38 (23-63)                                                  | 488 (443-559)            | 41 (21-71)                                             |  |  |  |
| $_{\sim}$   | 9                                | 340 (259–403)            | 30 (17-39)                                                  | 397 (362-446)            | 38 (26-58)                                             |  |  |  |

**Table 3.** Point density and average mean absolute error (MAE) for each ALS dataset for Skin Gulch and Hill Gulch, respectively. MAE was determined by the elevation difference between total station and RTK-GNSS survey points and interpolated ALS points.

| ALC dataset                       | Skin Gulch | l        | Hill Gulch                          |          |  |  |  |
|-----------------------------------|------------|----------|-------------------------------------|----------|--|--|--|
| Point density (pts/m 2 |            | MAE (cm) | Point density (pts/m 2 ) | MAE (cm) |  |  |  |
| 201210                            | 1.16       | 12       | 1.18                                | 23       |  |  |  |
| 201307                            | 2.00       | 11       | 2.21                                | 15       |  |  |  |
| 201310                            | 3.01       | 11       | 2.78                                | 9        |  |  |  |
| 201409                            | 3.27       | 12       | 3.82                                | 10       |  |  |  |
| 201506                            | 3.67       | 13       | 2.21                                | 13       |  |  |  |